# Methylome evolution suggests lineage-dependent selection in the gastric pathogen *Helicobacter pylori*

Florent Ailloud [1,2 ✉], Wilhelm Gottschall[1] & Sebastian Suerbaum [1,2 ✉]

The bacterial pathogen *Helicobacter pylori*, the leading cause of gastric cancer, is genetically highly diverse and harbours a large and variable portfolio of restriction-modification systems. Our understanding of the evolution and function of DNA methylation in bacteria is limited. Here, we performed a comprehensive analysis of the methylome diversity in *H. pylori*, using a dataset of 541 genomes that included all known phylogeographic populations. The frequency of 96 methyltransferases and the abundance of their cognate recognition sequences were strongly influenced by phylogeographic structure and were inter-correlated, positively or negatively, for 20% of type II methyltransferases. Low density motifs were more likely to be affected by natural selection, as reflected by higher genomic instability and compositional bias. Importantly, direct correlation implied that methylation patterns can be actively enriched by positive selection and suggests that specific sites have important functions in methylation-dependent phenotypes. Finally, we identified lineage-specific selective pressures modulating the contraction and expansion of the motif ACGT, revealing that the genetic load of methylation could be dependent on local ecological factors. Taken together, natural selection may shape both the abundance and distribution of methyltransferases and their specific recognition sequences, likely permitting a fine-tuning of genome-encoded functions not achievable by genetic variation alone.

[1] Medical Microbiology and Hospital Epidemiology, Max von Pettenkofer Institute, Faculty of Medicine, LMU Munich, Munich, Germany. [2] German Center for Infection Research (DZIF), Partner Site Munich, Munich, Germany. ✉email: ailloud@mvp.lmu.de; suerbaum@mvp.lmu.de

Methylation of DNA is a common epigenetic marker found in nearly all bacteria[1]. It involves the transfer of a methyl group from S-adenosyl-methionine to different positions of the DNA molecule by a DNA methyltransferase (MTase). In bacterial genomes, N6-methyl-adenine ($m^6A$), C5-methyl-cytosine ($m^5C$), and N4-methyl-cytosine ($m^4C$) modifications can be observed. Methyltransferases are often parts of restriction-modification (RM) systems. Presently, four types of RM systems have been described in bacteria[1,2]. Type I RM systems are multimeric enzymes with separate restriction, methylation, and specificity subunits[3]. Type II RM systems have separate restriction endonuclease and methyltransferase enzymes, with the exception of type IIG systems where both activities are either performed by a single protein or with the help of an additional specificity subunit[4]. Type III RM systems also have distinct restriction endonuclease and methyltransferase enzymes, but the endonuclease needs to bind the methyltransferase first in order to be active[5]. Type IV RM systems do not contain a methyltransferase and, unlike the other systems, restrict methylated DNA[6].

*Helicobacter pylori* is responsible for one of the most prevalent bacterial infections worldwide, affecting more than one-half of the human population[7,8]. It typically leads to chronic active gastritis, which can progress to further complications, such as peptic ulcers, MALT lymphoma, or gastric adenocarcinoma[9]. *H. pylori* is characterized by extensive inter-strain diversity which is the product of a high mutation rate, frequent recombination due to natural transformation, and a large and diverse repertoire of RM systems[10]. Such diversity is thought to be critical to *H. pylori*'s lifelong persistence and exceptional aptitude to adapt to the gastric environment and to evade the host immune responses[11]. To date, various functions have been attached to RM systems and methylation in bacteria[10]. In addition to its central role in distinguishing self from non-self DNA as part of the defense against phages, methylation has also been connected to transcriptional regulation, chromosome replication, stress response, antibiotic resistance, and virulence[12]. In *H. pylori*, several different methyltransferases have been associated with the regulation of gene expression[13–16]. In particular, the M.Hpy99III methyltransferase targeting the GCGC motif has been shown to influence cell morphology, expression of outer membrane proteins, and copper resistance[15]. Nevertheless, the transcriptomic and phenotypic effects associated with GCGC methylation were highly variable between strains, suggesting that genetic background plays a central role in determining the outcome of DNA methylation. The functions of other methyltransferases in *H. pylori* have only been assessed in single strains, and thus strain-specific effects could not be estimated.

On average, two to three RM systems are found in prokaryotic genomes[17,18]. In striking contrast, over 30 can be observed in a given *H. pylori* genome[19]. Only very few methyltransferases belong to the core genome and are strictly conserved in *H. pylori*[15,20]. Accordingly, the majority of methyltransferases are only found in subgroups of strains and thus belong to the accessory genome. This results in a very large number of possible combinations of RM systems, and a highly diverse methylome between strains[21–25]. The variability of RM systems in bacteria is a combination of different mechanisms of horizontal transfer. The target recognition domains (TRD) of type I and type III RM systems can be swapped via recombination to generate new specificities[23,26–28]. Alternatively, complete type II RM systems can be gained and lost by horizontal gene transfer[17,29]. In *H. pylori*, the global frequency and phylogenetic distribution of known RM systems as well as the influence of horizontal transfer have not yet been characterized exhaustively.

Methylation patterns are a combination of many individually methylated target motifs. Across the genome, single methylated motifs located in promoters, coding sequences or translation start sites have been associated with regulation of gene expression in *H. pylori*[14,15,30]. Consequently, changes in the position or frequency of motifs within methylation patterns have the potential to dramatically alter the effects of methyltransferases. Considering the genetic diversity of *H. pylori*, methylation patterns are likely to be substantially different between strains and lineages but such variability has not been investigated yet.

We propose that the methylome, the combination of a diverse repertoire of methyltransferases and a variable distribution of target sequences, represents an entire complex layer of (epigenetic) diversity, distinct yet intertwined with the nucleotide sequence (genetic) diversity of *H. pylori*. Furthermore, the phenotypes that have so far been associated with methylation in *H. pylori* suggest that this layer could contribute to rapid phenotypic diversification and adaptation to the ever-changing gastric environment. To determine the potential contribution of each methyltransferase to epigenetic diversity, we characterized the distribution of methyltransferases and the genomic patterns of methylated target motifs in *H. pylori*. Using a large collection of genomes representative of the geographical diversity of *H. pylori*, we show that type II RM systems are the most conserved in this species and that type II motifs are differentially affected by natural selection. In particular, we detected positive or negative correlations of motif density with the frequency of several type II methyltransferases across phylogeographic populations suggesting some direct evolutionary interplay between RM systems and methylation patterns. Finally, we reconstructed the complex evolution of the ACGT motif targeted by the M.Hpy99XI enzyme and characterized the striking contraction and expansion of this motif following geographically specific environmental factors.

## Results

**Systematic analysis of the diversity of 96 methyltransferases in *H. pylori*.** To quantify the variability of methyltransferases in *H. pylori*, we analyzed the distribution of 96 genes related to target-sequence specificity of RM systems (Table 1; Supplementary Data 1) in a collection of 541 genomes representative of the worldwide diversity of *H. pylori* (Supplementary Data 2). Within this collection, we sequenced 32 new *H. pylori* strains from the lesser characterized hpNEAfrica population and 31 additional strains from the hpAfrica1 population.

Most active type I, IIG, and III methyltransferases or specificity subunits were only detected in a small fraction of *H. pylori* genomes, with an average frequency below 5% (Table 1, Fig. 1a; Supplementary Data 3). Type II methyltransferases were by far the most widespread in *H. pylori*. In particular, a subset of ten type II genes was observed in more than 75% of the genome collection. Only two MTases, M.Hpy99III and M.HpyI were present in all genomes. M.Hpy99III targets the motif GCGC and

**Table 1 Distribution of 96 *H. pylori* RM systems in a globally representative collection of 541 *H. pylori* genomes.**

| RM system type | Total | Mean frequency (%) | Min–Max frequency (%)[a] |
|---|---|---|---|
| Type I | 47 | 3.0 | 0.2–12.8 |
| Type II | 31 | 25.3 | 1.1–100 |
| Type IIG | 13 | 4.7 | 1.5–12.6 |
| Type III | 5 | 2.2 | 0.2–5.5 |

[a]Min–Max frequency indicates the least and most frequent RM systems of a specific type.

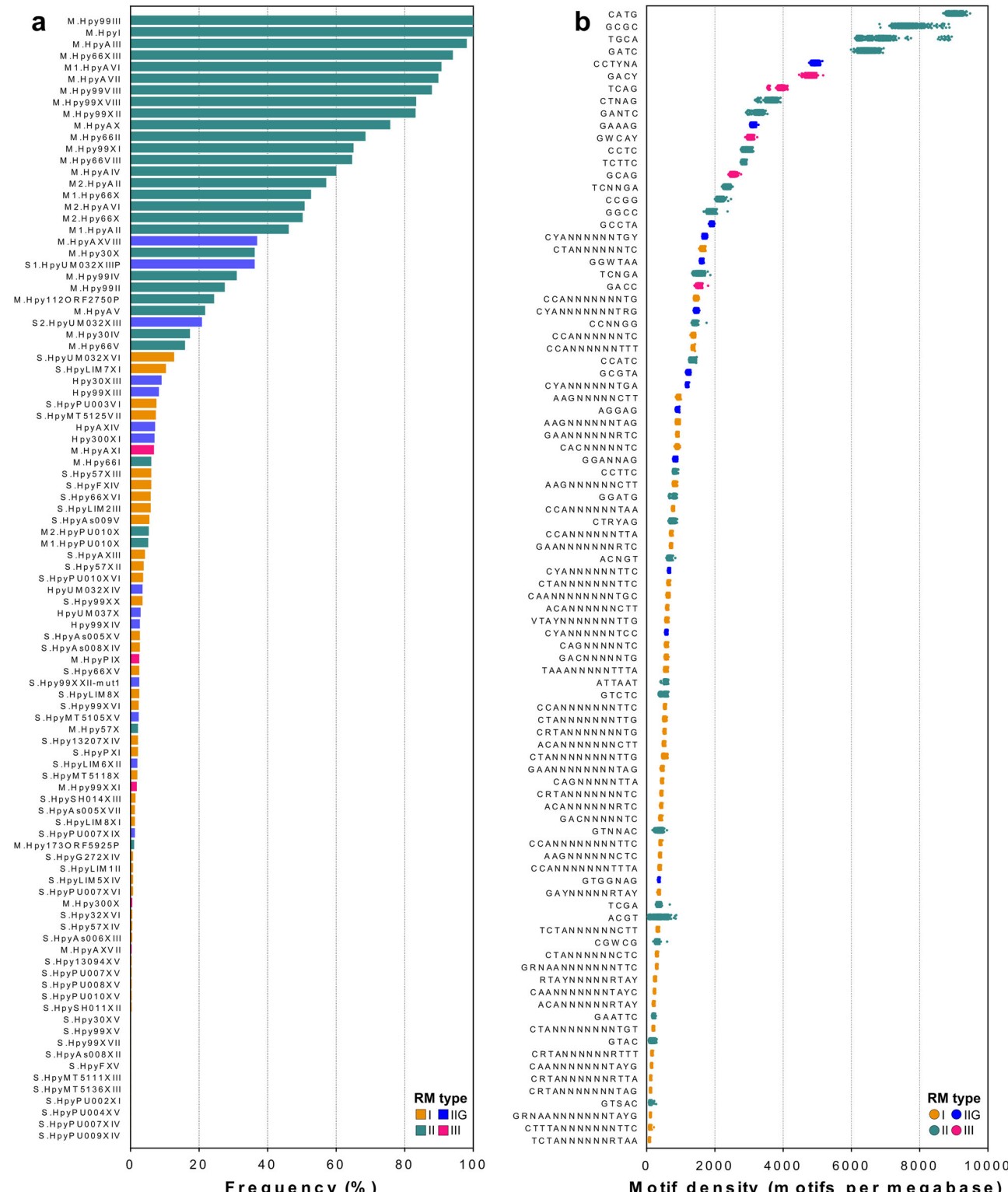

**Fig. 1 Distribution of methyltransferases, target-sequence specificity subunits, and target motifs in _H. pylori_. a** The frequency of 96 genes related to target-sequence specificity of RM systems was calculated across a collection of 541 _H. pylori_ genomes. The gene names are indicated on the _y_-axis. Bars are colored according to the R-M system type. **b** The frequency of 92 target motifs was calculated across the same collection of _H. pylori_ genomes. The motif sequences are indicated on the _y_-axis. Each dot represents a single genome and is colored according to R-M system type.

has been associated with the regulation of gene expression[15], while M.HpyI targets CATG motifs and is part of the _iceA_ RM system, a potential marker for _H. pylori_ strains associated with gastric carcinoma[31,32]. On the opposite of the frequency spectrum, nine type II methyltransferases were detected in less

than 25% of genomes. This includes the paired M1.HpyPU010X and M2.HpyPU010X methyltransferases, which methylate the exact same target-sequence motif GGATG and are part of the rare group of RM systems regulated by a controller subunit in _H. pylori_[33,34].

**Table 2 Direct flanking repeats in six type II RM systems from *H. pylori*.**

| MTase | Repeat length (bp) | Inter-strain identity (%) | Intra-strain identity (%) | Prevalence in MTases (%) |
|---|---|---|---|---|
| M.Hpy99XI | 95 | 86.1 | 86.2 | 67.0 |
| M.Hpy66II | 115 | 97.1 | 98.4 | 2.5 |
| M.Hpy99IV | 395 | 94.5 | 97.9 | 41.7 |
| M.Hpy66I | 376 | 93.1 | 97.8 | 21.2 |
| M1.HpyAII/M2.HpyAII | 65 | 92.9 | 90.9 | 59.6 |
| M.HpyAIV | 118 | 95.5 | 93.3 | 65.8 |

On average, type I specificity subunits and type II methyltransferases were more conserved (95.9-96.4%) than type IIG and III methyltransferases (91.9–92.8%) (Supplementary Fig. 1a, b). One of the main differences between the type II RM systems and the others is the localization of the target recognition domain (TRD). In type I and a subset of type IIG systems, the TRDs are located in specificity subunits, detached from the endonuclease and methyltransferases genes. Within the same specificity subunit allelic backbone, distinct target sequences can be obtained by recombination between different TRDs. Using phylogenetic and consensus analysis, we were able to group the TRDs from 47 type I S subunits in only five allelic backbones (Supplementary Fig. 2a). Likewise, the five type IIG S subunits actually shared a single backbone (Supplementary Fig. 2b). In type III and a different subset of type IIG systems, the TRDs are located in the methyltransferase. Similar to S subunits, the TRDs in those systems can recombine within similar allelic backbones. We identified three backbones among seven TRDs for type IIG methyltransferases (Supplementary Fig. 2c) and three backbones among five TRDs for type III methyltransferases (Supplementary Fig. 2d). In particular, one type III backbone corresponded to the previously characterized *modH* methyltransferase for which 14 TRDs have been identified so far, although a majority have not been associated with a target sequence yet[35,36]. Consequently, the low frequency of type I, IIG, and III methyltransferases or specific subunits is likely the result of competition for the limited amount of allelic backbones available in *H. pylori*.

In type II systems, the TRDs are not able to recombine and thus methyltransferase loci are associated with a single target sequence and do not compete with each other. Instead, the diversity of type II RM systems is typically based on the gain and loss of specific systems. By examining the local context of type II gene clusters, we identified six type II systems flanked by direct repeats that can potentially lead to spontaneous deletion events (Table 2; Supplementary Data 4)[37–39]. These results were further supported by the observation of a single copy of the repeats at similar genomic locations in strains where these RM systems are absent (Supplementary Fig. 3). Flanking repeats displayed both high inter- and intra-strain identity supporting the possibility of intramolecular recombination. Interestingly, repeats were not systematically found in all alleles of each affected methyltransferase (Table 2). Other type II systems did not display similar repeats and thus are likely gained or lost only through natural transformation. Overall, the frequency of type II methyltransferases was moderately correlated with their nucleotide identity, suggesting that the less prevalent enzymes were simply acquired more recently (Supplementary Fig. 1c). As *H. pylori* is considered to be constitutively competent[40,41], such RM systems could still be transferred by homologous recombination following natural transformation.

**Methylation patterns follow different evolutionary trajectories**. Variability in the frequency of target motifs has the potential to affect the role of MTases through changes in methylation patterns across the genome of *H. pylori*. Therefore, we determined the frequency of 92 target motifs in our *H. pylori* genome collection. To account for differences in genome length, the total number of motifs for each strain was normalized by the length of its genome and scaled to obtain the density in motifs per megabase (Fig. 1b). The results spread over a 100-fold range of frequencies with the lowest motif density calculated for TCTANNNNNNRTAA (84 motifs/Mb), methylated by a type I system, and the highest obtained for CATG (9036 motifs/Mb), methylated by a type II system. A similar pattern was observed across the whole dataset, with low densities associated with type I motifs and higher densities associated with type II motifs. Intriguingly, the two target sequences with the highest motif densities, CATG and GCGC, are recognized by the two only RM systems whose MTases are fully conserved in *H. pylori* (Fig. 1b). Additionally, we compared the motif frequencies we obtained from whole genomes to ones obtained from a core gene alignment (Supplementary Fig. 4). Seven motifs showed a frequency increase of >25% in the whole versus core comparisons, including the type II motifs ATTAAT and TCGA. However, these motifs have a fairly low frequency overall (min: 91 motif/Mb, max 915 motif /Mb) and thus the differences in absolute number of motifs were relatively small (92 motifs/Mb difference on average). Consequently, the frequency of the majority of motifs appears to be influenced by the evolution of the whole genome rather than the gain and loss of motifs through the accessory genome.

Next, we measured the dependence between motif densities and RM system frequency using the distance correlation method in order to detect non-linear associations (Fig. 2a). A moderate correlation was detected between the two variables (distance correlation measurement dCor=0.48; right-tailed permutation test with 1000 bootstraps $p = 0.009$), which suggests some interdependence between the function of RM systems and the density of their respective motifs. In particular, high-density motifs (>5000 motifs per Mb) were specifically associated with high frequency (>50%) type II RM systems.

The high mutation rate (i.e., approx. $10^{-5}$ mutations per site per year) characterizing *H. pylori* has the potential to rapidly change methylation patterns[42–44]. Accordingly, we assessed the genetic variability of each motif across the species. Based on a core-genome alignment built from our collection of 541 *H. pylori* genomes, we calculated the average number of motifs shared between genomes and scaled it to the mean number of motifs per genome to determine the stability of each motif pattern (Fig. 2b). The stability of target motifs in *H. pylori* ranged from 25 to 88% with an overall mean of 70%. A moderate correlation was observed between motif stability and motif density (dCor = 0.46, $p = 0.0009$). Interestingly, only motifs methylated by type II RM systems and with very low density (<500 motifs per Mb) displayed a stability below 50%, suggesting that the genomic patterns of these motifs carry a higher genetic load than type II motifs with a higher density (>5000 motifs per Mb) which, in contrast, systematically had a stability above 70%.

In order to look for further evidence of selection pressures on methylation patterns, we calculated the expected frequencies of

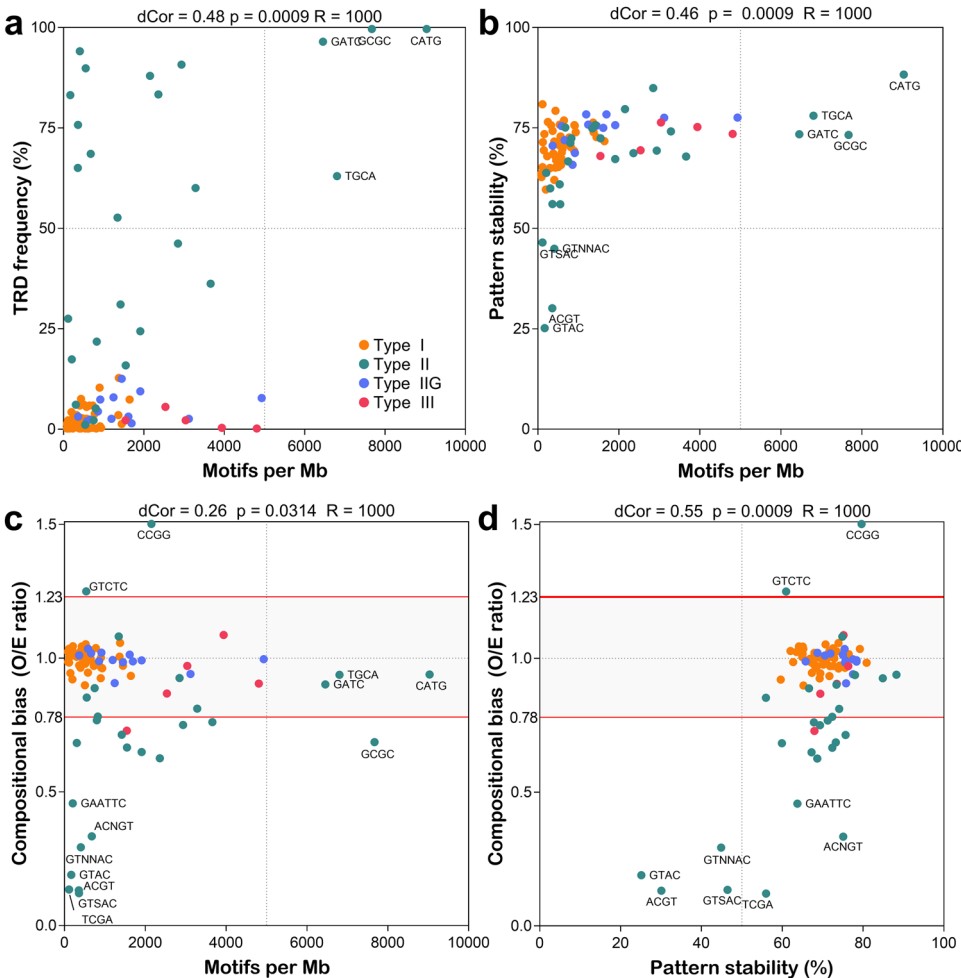

**Fig. 2 Interaction between methylome attributes.** The correlation between different methylome variables was measured by distance correlation analysis with 1000 bootstrap replicates (R). *p*-values and the distance correlation coefficients are indicated above each scatter plot. **a** Motif density ~ TRD frequency **b** Motif density ~ Pattern stability. **c** Motif density ~ Compositional bias. **d** Pattern stability ~ Compositional bias. Dots are colored according to the RM system enzymatic type. Selected data points are annotated with the corresponding target motif. Cut-offs for under- (<0.78) and over- (>1.23) represented are indicated by red horizontal lines for compositional bias data.

target motifs using probabilistic models based on the nucleotide composition of *H. pylori*[45,46]. By comparing expected and observed frequencies (i.e., compositional bias CB), we determined which motifs were either under- (CB < 0.78) or over-represented (CB > 1.23) and thus potentially under selection (Fig. 2c). As shown in other prokaryotic genomes[18], 22% of the target motifs of type II R-M systems were strongly under-represented with a compositional bias below 0.5. In contrast, one type II motif, CCGG, was over-represented with a compositional bias above 1.5. Additionally, we replicated the compositional bias calculations using two alternative methods and confirmed these observations (Supplementary Fig. 5). Compositional bias did not appear to be a good predictor of motif density and only a weak correlation was observed between these variables (dCor = 0.26, *p* = 0.03). Nevertheless, high-density type II target sequences only displayed a limited amount of under-representation confirming that those motifs are genetically maintained within the species. Compositional bias was, however, strongly associated with pattern stability (dCor = 0.55, *p* = 0.0009). This correlation suggests that the extremely high instability of some motif patterns is likely the result of natural selection pressures that ultimately lead to the removal and an overall under-representation of the motif (Fig. 2d).

**Type II methyltransferases have direct positive or negative selective effects on methylation patterns**. *H. pylori* is known to exhibit phylogeographic patterns reflecting the ones of its human host, owing to their long co-evolutionary association[47]. Phylogeographic populations of *H. pylori* are genetically distinct from each other, with the most well-known example being the virulence-associated *cag* pathogenicity island (*cag*PAI). For instance, the *cag*PAI displays phenotypical variation between Eastern and Western strains in the first super-lineage of *H. pylori*, while being completely absent in the second super-lineage at the origin of the hpAfrica2 population[48,49]. Epigenetic variations across phylogeographic populations of *H. pylori* have not been studied in comparable detail. Accordingly, we performed clustering analysis to examine the distribution heterogeneity of 31 type II methyltransferases in seven major phylogeographic populations of *H. pylori* (Fig. 3).

The clustering of populations according to their frequency patterns of methyltransferases (Fig. 3) mimicked the phylogeny of *H. pylori*[50]. This suggests that human host migration and geographic isolation contributed to the variability of type II RM systems in *H. pylori*[51]. Furthermore, the clustering of methyltransferases according to their frequencies in *H. pylori* populations (Fig. 3) revealed three distinct clusters: (i) a cluster of 11

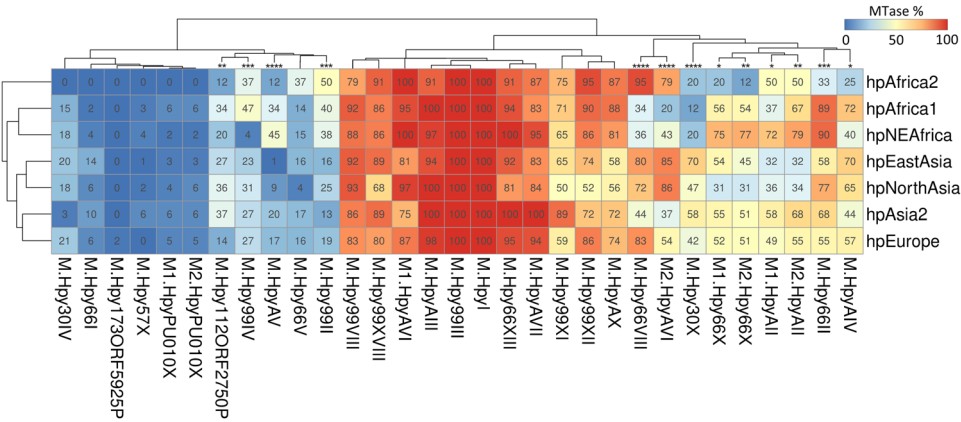

**Fig. 3 Geographical variation of type II methyltransferase frequency in *H. pylori*.** The frequency of 31 methyltransferases was calculated in 7 geographical populations of *H. pylori* within a collection of 541 genomes. The heatmap is color-coded from blue to red according to the proportion indicated in each cell and is clustered in both axes. Methyltransferases with significant variation between populations are indicated by asterisks (Chi-square *p*-value, * <0.05, ** <0.01, *** <0.001, ****<0.0001).

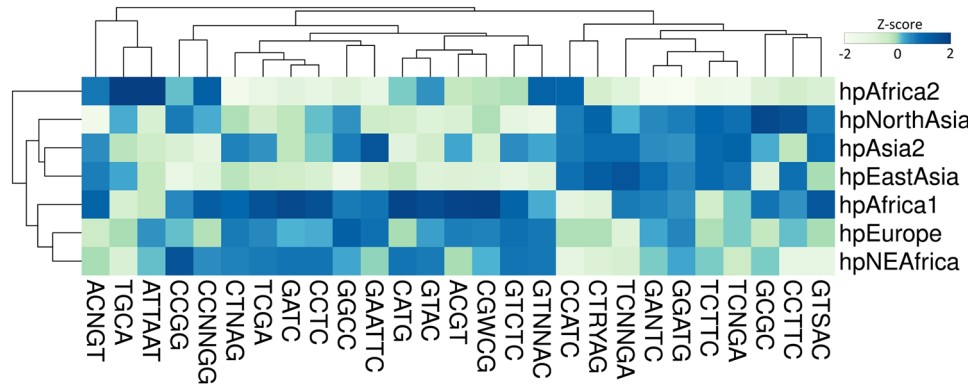

**Fig. 4 Geographical variation of type II target motif frequency in *H. pylori*.** The density of 27 target motifs (motifs/Mb) was calculated in 7 geographical populations of *H. pylori* within a collection of 541 genomes. The heatmap is color-coded according to the Z-score, representing the standard deviation calculated for each motif separately.

"common" enzymes with frequencies ranging from 50 to 100% depending on the phylogeographic population, (ii) a cluster of 11 "rare" enzymes below 50% frequency in every population, and (iii) a cluster of nine "variable" enzymes displaying large variations of frequency across all populations.

We detected significant variation of frequencies between phylogeographic populations of *H. pylori* for 13 methyltransferases (Chi-square test, *p* < 0.05), suggesting that these genes may be affected by positive or negative selection in specific lineages. In the cluster of "rare" enzymes, several methyltransferases, M.HpyAV (C^m5^CTTC/GA^m6^AGG), M.Hpy99II (GTS^m6^AC) and M.Hpy99IV (^m4^CCNNGG) were overall more frequent in distinct African populations than in Asian populations. In the cluster of "variable" enzymes, M.Hpy66VIII (TGC^m6^A) and M2.HpyAVI (^m5^CCTC) displayed similar patterns across phylogeographic populations. Both were strongly associated with hpEastAsia and hpNorthAsia as well as with hpAfrica2, but were far less common in the other African populations, hpAfrica1 and hpNEAfrica. Additionally, M.Hpy30X (^m4^CTNAG) appeared specifically depleted in all African populations, while M.Hpy66II (A^m4^CNGT) was clearly enriched in the hpAfrica1 and close relative hpNEAfrica populations.

Next, we asked whether motif densities would behave in a comparable way and performed a similar analysis (Fig. 4). Surprisingly, every motif showed a significant variation of density between populations (Kruskal–Wallis test, *p* < 0.05). Because

methylation patterns are intertwined with the genome, they are similarly affected by genetic drift and natural selection. Therefore, differences are expected when comparing divergent lineages. Nevertheless, many motifs displayed comparable patterns of density. The largest cluster was composed of motifs with higher densities in hpAfrica1 and related hpNEAfrica and hpEurope populations, while the second largest cluster contained motifs with increased densities in all Asian lineages. On the contrary, only a small number of motifs appeared to have expanded in the hpAfrica2 population. Overall, the largest absolute variation of density was observed for the TGCA motif with ~3000 motifs in hpAfrica1 and ~4500 motifs in hpAfrica2. On a relative scale, the greatest variation was observed for the ACGT motif with ~50 motifs in hpEastAsia and ~400 motifs in hpAfrica1, representing an 8-fold difference.

Finally, we sought to investigate the relationship between methyltransferase and target motif density across phylogeographic populations in order to determine if methylation can directly influence the evolution of motif patterns. Consequently, we used logistic regression to investigate how the frequency of methyltransferases affects the motif density in phylogeographic populations. We found significant positive and negative interactions for four and three methyltransferases, respectively (Supplementary Data 5). In this context, a positive interaction indicates that the motif density increases as the methyltransferase frequency increases too while a negative interaction

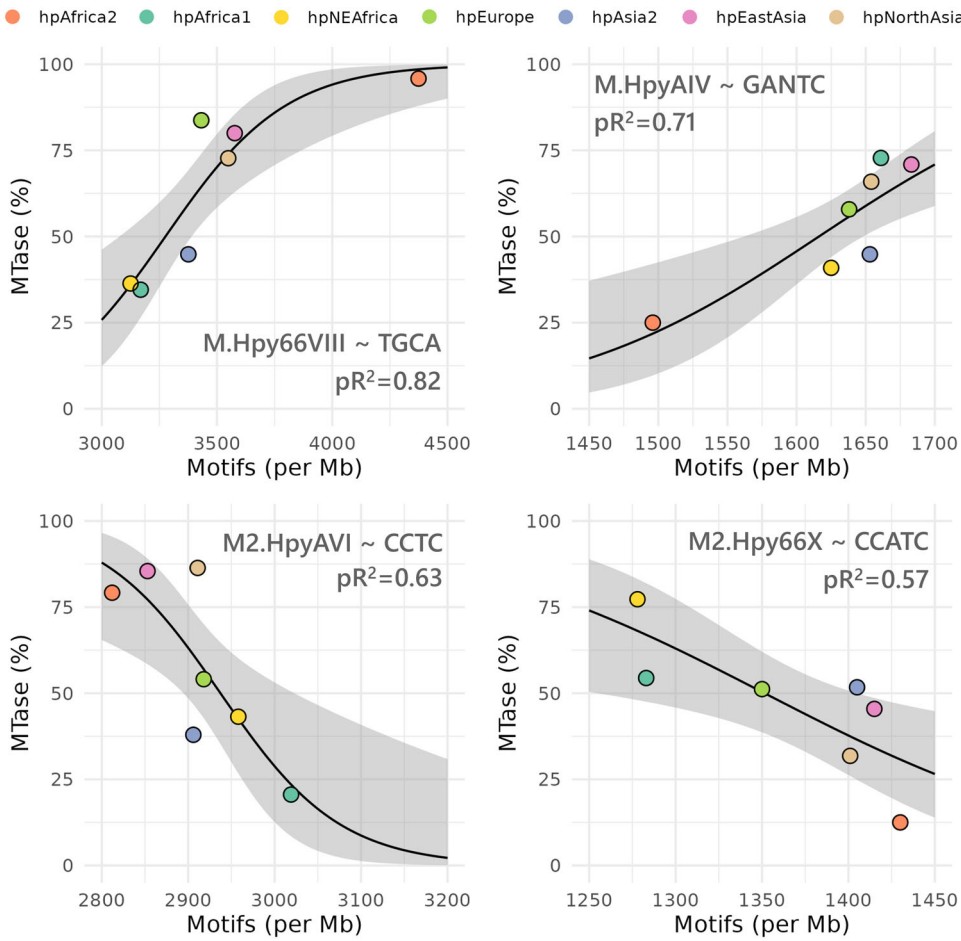

**Fig. 5 Positive and negative interaction of type II methyltransferases with motif patterns.** The interactions between methyltransferases and motifs were estimated using a generalized linear model with a quasibinomial distribution and a logit link function. A 95% confidence interval is indicated by a gray ribbon. Goodness-of-it was assessed using a chi-square test ($p < 0.05$). Pseudo-$R^2$ (McFadden) was calculated using the ratio of residual deviance over the null deviance and is indicated on the plots.

indicates the opposite. Two examples of each type of interaction are displayed in Fig. 5. Interestingly, the hpAfrica1 and hpAfrica2 populations were at the opposite ends of the spectrum for each interaction. Notably, the motif density of TGCA increased from ~3000 to 4500 with the frequency of the cognate methyltransferase M.Hpy66VIII going from ~30 to ~100%. In contrast, an increase of ~50% of the M2.HpyAVI frequency was associated with a decrease from ~3000 to 2800 in the CCTC motif density.

The presence of the cognate restriction endonuclease in type II RM systems has been shown to lead to the avoidance of palindromic motifs in several bacterial species[52–54]. The Hpy99III (GCGC) and HpyI (CATG) type II systems are known to be facultatively lacking an endonuclease (i.e., orphan methyltransferases) in *H. pylori*[15,55]. In the same way as for methyltransferases, we tested the relationship between the frequency of the endonuclease and the motif density for these two RM systems but did not find any significant interaction (Supplementary Fig. 6).

These results suggest a direct selective effect of methylation on individually methylated motifs. Furthermore, the existence of both positive and negative interactions indicates that the evolution of motif patterns is highly specific to each methyltransferase and implies that distinct RM systems might fulfill specific functions with overarching effects on the fitness of *H. pylori*.

**Lineage-specific expansion and contraction of the type II m5c motif ACGT.** Among type II motifs, the ACGT motif displayed the largest relative variation of density between phylogeographic populations of *H. pylori*. However, these differences were not correlated with the frequencies of the cognate methyltransferase across populations (Supplementary Data 3). For instance, the Hpy99XI methyltransferase was present in 65% of the hpEastAsia strains which only contained around 53 ACGT motifs/Mb, but was observed in 71% of the hpAfrica1 genomes which contained 295 motifs/Mb (Supplementary Fig. 7). In our global analysis of methylated motifs in *H. pylori*, the ACGT target sequence was also one of the most highly unstable and under-represented motifs (Fig. 2). In order to determine the underlying causes for the peculiar evolution of ACGT, we first compared the genomic patterns of this motif to determine the overlap across four representative phylogeographic populations (hpAfrica1, hpAfrica2, hpAsia2, and hpEastAsia). A minimal number of motifs were shared across all strains and most motifs were completely specific to each population (Fig. 6a). This result echoes the low pattern stability observed previously and suggests that the ACGT motif underwent rapid evolution. Intriguingly, the proportion of motifs shared between populations did not quite reflect the evolutionary history of *H. pylori*. For instance, the number of motifs shared between hpAfrica1 and hpAfrica2 was higher than between hpAfrica1 and hpEastAsia. At the same time, the number of motifs specific to each population was also highly variable

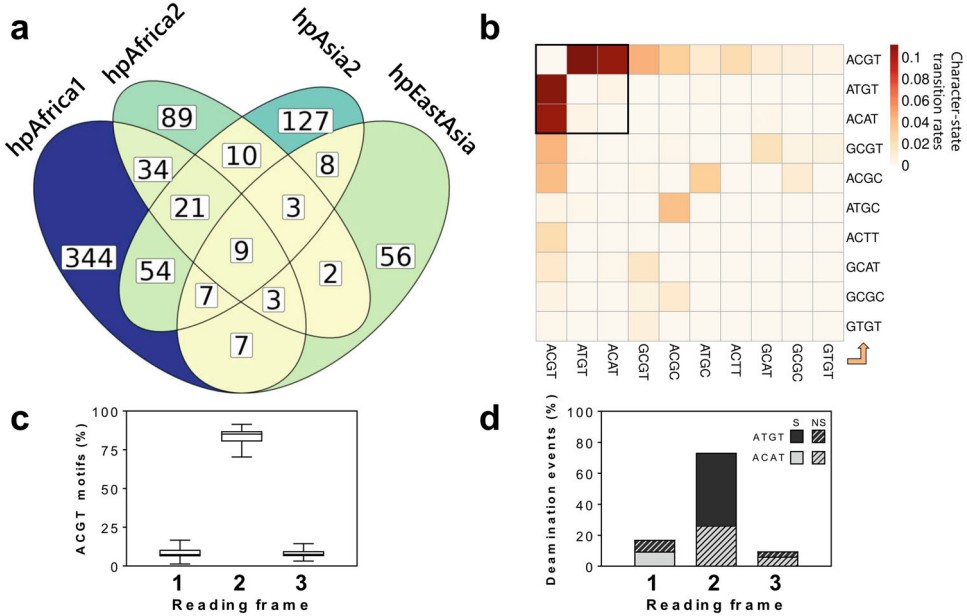

**Fig. 6 Genomic patterns of the ACGT target motif. a** Overlap of motif patterns between representative populations of *H. pylori*. All genomes included in this analysis were compared in a pairwise fashion to determine how many motifs are present at the same genomic coordinates in each pair and the average between populations is presented. Sections of the Venn diagram are colored from yellow to blue according to the number of overlapping motifs. **b** Character-state transition rates between ACGT and its most frequent allelic variants. The direction of character-state change is displayed from the *x*-axis to the *y*-axis. **c** Distribution ACGT motifs located within coding sequences according to their reading frames and represented by a box plot. The median is indicated by the central line across the box. The lower and upper hinges represent the 25th and 75th percentile, respectively. The ends of the lower and upper whiskers represent the minimum and maximum data points, respectively. **d** Effect of state transition between ACGT and ATGT/ACAT according to the reading frame (Syn: synonymous; Non-syn: non-synonymous).

which suggests overall that the genomic patterns of the ACGT were shaped by very specific differences in local environments.

Next, we used a representative subset of ten genomes each from the four major phylogeographic populations of *H. pylori* to perform ancestral-state reconstruction analysis and understand how ACGT evolved during the divergence of these lineages. Based on the ACGT motif patterns observed in modern strains of *H. pylori*, the ancestral-state reconstruction method can identify the ancestral phylogenetic nodes in which individual motifs were either gained or lost. By combining the reconstruction of all ~7000 unique positions in which ACGT motifs were observed in our subset of 40 representative genomes, we determined the transition rates between ACGT motifs and its most frequent allelic variants (Fig. 6b). The transition matrix was heavily skewed toward the ATGT and ACAT variants. The M.Hpy99XI enzyme, targeting the ACGT motif, is a $^{m5}$C methyltransferase. Spontaneous deamination of 5-methylcytosine to thymine is known for making $^{m5}$C motifs more prone to mutations compared to $^{m6}$A or $^{m4}$C motifs[56]. Interestingly, in the case of the ACGT motif, deamination would produce either ATGT or ACAT variants (depending on the affected DNA strand), suggesting that deamination played a major role in the evolution of this motif. To understand the potential fitness cost of the ACGT motif mutations, we focussed next on motifs located within coding sequences. In particular, we determined in which reading frame those motifs are positioned and found that they are heavily biased towards the second frame (i.e., motif starting at the second base of codons) (Fig. 6c). Furthermore, the majority of deamination events resulted in synonymous mutations, with no effect on the encoded protein sequences (Fig. 6d). Mutations in the second frame were skewed toward synonymous ATGT variants, corresponding to deamination of ACGTs motif on the sense strand of coding sequences (Fig. 6d). These results indicate that deamination of methylated ACGT motifs would mostly result in

silent mutations and would likely not cause any secondary effects outside of the loss of methylation (and vice versa, gain of ACGT motifs from ATGT/ACAT would be mostly silent). Consequently, the evolution of the ACGT motif itself seems to be partially constrained by the purifying selection influencing coding sequences, affecting the rate at which each strand gets deaminated and minimizing its effect on protein-coding sequences. Accordingly, the main effect of ACGT changes is solely the loss of methylation markers and thus any natural selection pressure associated with the evolution of A$^{m5}$CGT motif patterns is presumably operating at the epigenetic level without being hindered by its effects at the genetic level.

Finally, we used our ancestral-state dataset to reconstruct the expansion and contraction of the ACGT motif patterns at each major ancestral node as well as at the root (Fig. 7). Our analysis indicates that ~130 ACGT motifs were present in the common ancestor of *H. pylori* at the root, dated at ca. 100,000 years according to previous studies[49]. From there, the motif pattern expanded at the hpAfrica1/Asian node whereas it contracted in the hpAfrica2 node mainly because of deamination. The expansion continued between the hpAfrica1/Asian node and the hpAfrica1 leading to a higher number of ACGT motifs observed in modern Africa1 strains. On the opposite, the motif patterns contracted between the hpAfrica1/Asian and the Asian nodes leading to a severe reduction in the number of motifs. Interestingly, the contraction seemingly slowed down in the hpAsia2 node but appeared to accelerate in the East-Asia node, explaining the unusually low number of motifs seen in the modern East-Asian *H. pylori* population. Overall, the population-specific trends of contraction and expansion across those genetically distinct geographical populations indicate that external factors, such as the host physiology or the local environment, are having a selective effect on the evolution of the ACGT methylation patterns. Furthermore, the strong positive selection

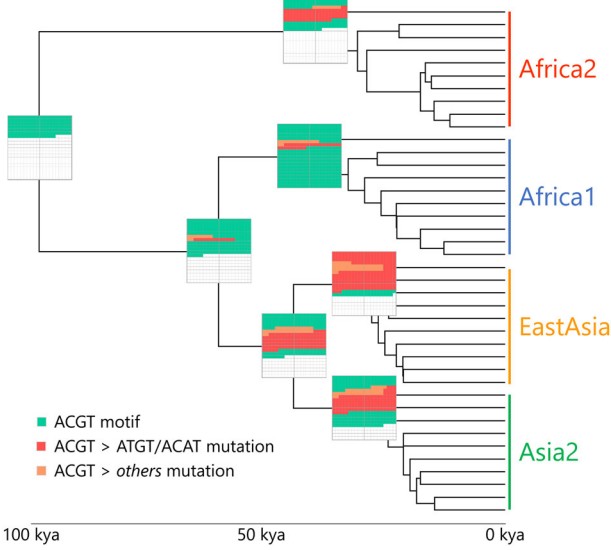

**Fig. 7 Ancestral reconstruction of the ACGT motif patterns across *H. pylori* major phylogeographic populations.** A time-scaled tree was created using 40 representative strains from the hpAfrica1, hpAfrica2, hpEastAsia, and hpAsia2 populations. The ancestral states for each genomic position displaying ACGT motifs in modern strains were first predicted separately by stochastic character mapping and then concatenated to generate the ancestral patterns. A graphical representation of the ACGT motif patterns is given by boxes located at each ancestral node. Green squares represent new individual ACGT motifs, while red and orange squares indicate ACGT motifs lost through ATGT/ACAT mutations and other mutations, respectively.

pressures which are likely required to expand and maintain the ACGT motifs in hpAfrica1, in contrast to the negative selection pressures needed to purge most of the motifs in hpEastAsia, would suggest the ACGT motif may have a role in the evolutionary fitness of *H. pylori*.

## Discussion

*H. pylori* is a bacterial pathogen with exceptionally high genetic diversity whose phylogenetic structure reflects the one of its host[10,47]. The long and intimate association between *H. pylori* and humans[47] has likely been sustained by the ability of this organism to rapidly adapt to the harsh environment characteristic of the gastric habitat[11]. The diversity of *H. pylori* is believed to be the main factor underlying its capacity for host adaptation. While the genetic variability of *H. pylori* has been extensively characterized, from its worldwide phylogeographic structure to its within-host diversity across stomach niches[10], the evolution of its methylome has not yet been investigated in the context of its global population structure. In this study, we undertook an in-depth characterization of the distribution of RM systems as well as the evolution of methylation patterns across the phylogenetic spectrum of *H. pylori*.

We started by investigating the frequency of the 96 RM systems characterized so far in *H. pylori* among a genetically diverse collection of 541 genomes. Type II methyltransferases were by far the most conserved in the species. This result is likely tied to their genetic organization and the separation of the endonuclease and methyltransferase activities. In particular, this organization can lead to post-segregational killing from residual endonuclease activity after the loss of type II RM systems[38,57]. Nevertheless, the two methyltransferases that are fully conserved in *H. pylori* are not always attached to an endonuclease (i.e., orphan

MTases)[15,32], indicating that these methyltransferases are likely maintained in the genome of *H. pylori* because they provide an evolutionary benefit. Furthermore, for both systems, the presence of the endonuclease did not appear to influence methylation patterns. On the other hand, the frequency of other types of RM systems (I, IIG, and III) was strongly limited by the fact that multiple TRDs have to compete for a limited number of methyltransferase or specificity subunit allelic backbones. The specificity of type I RM systems is determined by two TRDs located in the S subunits[23,58]. The shuffling of TRDs via recombination can produce an almost infinite number of target motifs[28,59], which is reflected by the large number of type I motifs found in *H. pylori*. For type IIG and type III RM systems, the specificity is determined by single TRDs, which can also be transferred between allelic backbones. Intriguingly, only a single backbone was identified among all the type IIG RM systems controlled by a specificity subunit. The most variable type III RM system in *H. pylori* is *modH* with 17 distinct TRD identified so far[35]. However, the target motifs have only been characterized for three TRDs of *modH*, indicating that type III systems need to be characterized further.

The target motifs methylated by type II methyltransferases were overall more frequent than the ones from other types of RM systems. Globally, the frequency of methyltransferases was only moderately correlated with the densities of their cognate target sequences. As a notable exception, the only two universally conserved *H. pylori* MTases recognize the two by far most abundant motifs (GCGC and CATG). Similarly, high-density type II motifs, such as CATG and GCGC, were characterized by high pattern stability and limited compositional bias. Both the MTases M.HpyI (C$^{m6}$ATG) and M.Hpy99III (G$^{m5}$CGC) have been shown to influence the expression of many genes in *H. pylori*[15,16]. The regulation of gene expression via methylation has also been shown for other *H. pylori* methyltransferases[13,14,16], as well as in other bacterial species[60–62]. While several studies have pointed out a potential role of target motifs within promoter elements or coding sequences[14–16,30], the transcriptional mechanisms have not been clearly characterized yet. Moreover, the strain specificity of gene regulation observed in these studies suggests a role of variable methylation patterns. On the opposite, the under-representation of multiple type II motifs in the genome of *H. pylori* was strongly associated with the lower stability of their motif patterns. Consequently, these highly unstable motifs are likely under selection pressures that gradually remove them from the species. Overall, the large variation in compositional bias and pattern stability among target motifs indicates that natural selection pressures do not affect the methylome globally but are more likely specific for each methyltransferase and cognate motif, suggesting these serve different functions in the biology of *H. pylori*. Motif avoidance is a phenomenon known for causing the depletion of restriction sites in bacterial genomes in order to limit self-restriction while still maintaining active endonucleases required for phage defense[45,52–54,63]. Interestingly, prophages in *H. pylori* also display a phylogeographic structure[64–66]. This effect is typically more pronounced in type II RM systems, which is likely due to their fixed TRD and higher diversity compared to other types[63]. Despite the presence of many cognate endonucleases, type II motifs are obviously not all affected similarly by motif avoidance. This could be explained by target motifs having distinct susceptibilities to self-restriction, or, the existence of additional selection pressures on specific motifs, counter-acting motif avoidance. Furthermore, context-dependent mutations may also be responsible for the instability of some motifs. For example, 5-methylcytosine is prone to spontaneous deamination and thus typically displays an increased mutation rate. Nevertheless, m5c motifs show large differences in terms of compositional bias

which also suggests that specific motifs are either maintained or lost via additional factors.

As the evolution of *H. pylori* is intimately linked to human migrations and geographical isolation, we investigated the distribution of type II methyltransferases and target motifs among the phylogeographic populations of *H. pylori*. Our analysis revealed a subset of nine methyltransferases showing large variations of frequency between populations. Four of these methyltransferases are flanked by direct repeats, which most likely contribute to their higher variability. While the frequency of methyltransferases was likely heterogeneous between ancestral populations of *H. pylori* due to founder effects following human migration events, the distribution of these methyltransferases in modern panmictic populations of *H. pylori* is either explained by either pure genetic drift or geographically dependent fitness effects. Subsequently, we identified correlations between methyltransferase frequency and motif density for seven type II RM systems, suggesting that methylation can indeed shape motif patterns via natural selection. Interestingly, we observed both positive and negative correlations. Negative correlations indicate that the presence of the methyltransferase leads to the elimination of its cognate motif. In addition, selection pressure leading to the depletion or enrichment of motifs might also be driven by the host immune system[67,68]. By contrast, positive correlations imply that the presence of the methyltransferase leads to a positive selection of its cognate motif. Direct selective effects on methylation patterns leading to the enrichment and/or maintenance of a motif in the genome have not been described and suggest that specific methylation patterns can contribute to the evolutionary success of *H. pylori*. This effect was particularly evident for the M.Hpy66VIII methyltransferase targeting the motif TGCA. In this case, the gradient of methyltransferase frequency and motif density distinguished the hpAfrica1/hpNEAfrica, hpEastAsia/hpNorthAsia, and hpAfrica2 populations. These specific groups of populations are highly divergent from each other. In particular, the CagA virulence factor is functionally distinct in Western versus Eastern populations of *H. pylori* while the *cag*PAI T4SS is completely absent in hpAfrica2 since it descends from a separate super-lineage than the other populations. To date, the role of M.Hpy66VIII has not been investigated but the maintenance of this methyltransferase and TGCA motif patterns in hpAfrica2 is likely related to specific local environmental factors.

In specific cases, local environmental factors may have selective effects on methylation patterns leading to geographical variation, independently of the frequency of the methyltransferase. Our regression analysis suggests that fluctuations in MTase frequency could only account for differences in motif densities in ~20% of the cases. In particular, the ACGT motif displayed the highest relative change in density across phylogeographic populations but showed no correlation with methyltransferase frequency. Demographic bottlenecks and the rapid evolution of *H. pylori* following repeated human migration events[69–71] most likely precipitated the evolution of the ACGT methylation pattern in the species. We hypothesize that the striking difference in the evolution of the ACGT motif between the hpAfrica1 and hpAfrica2 populations could be related to the acquisition of the *cag* pathogenicity island in the former[49]. The *cag*PAI is thought to provide a fitness advantage to *H. pylori* and to have contributed to the spread of hpAfrica1 through Africa and subsequently to other regions of the world[48]. How could the low density of the ACGT motif observed in Asian populations be explained in this scenario? Our analyses suggest that the deamination of 5-methylcytosine was one of the main drivers of motif depletion for ACGT. Since it is well established that Asian variants of *cag*PAI components, and CagA in particular, are associated with stronger inflammation and ultimately carcinogenicity[72–74], we speculate that increased host cell interaction and inflammatory response may have contributed to increased deamination and hence caused loss of ACGT motifs in Asian populations. Furthermore, because of the placement of ACGT motifs within coding sequences, transitions between ACGT and its deaminated variants are mostly silent, facilitating the evolvability of this motif. The evolution of the ACGT motif in the methylome context is thus strongly separated from its genomic context and thus any hypothetical selective effects involved in this process would be mainly driven by the epigenetic status of this motif rather than its genetic sequence. As speculated above, the various trajectories taken by the ACGT motif pattern from the common ancestor to the modern populations suggest that local environmental cues can greatly affect the genetic load of methylation and thus the epigenetic landscape of *H. pylori*. The fact that the GCGC motif was neither underrepresented nor unstable additionally points to the specificity of evolutionary pressures affecting 5-methylcytosines.

In conclusion, the methylome of *H. pylori* is a major contributor to its overall variability. Because the evolution of methylation patterns is constrained by their genetic sequence and the distribution of RM systems is influenced by their gene organization, the methylome is completely intertwined with the genetic variation of *H. pylori* and dependent on the phylogeographic structure of the species. Yet, the methylome is also shaped independently by selection pressures able to expand or contract motif patterns as a direct result of methylation, and environmental factors whose selective effects appear dependent on specific motifs and lineages. Third-generation sequencing technologies have permitted the rapid discovery of many new methyltransferases and the characterization of their target sequences in diverse bacterial species. Quantitative frameworks, such as the one expanded in this study, will contribute to the identification of methyltransferases whose functions extend beyond the standard phage defense model.

## Methods

**Construction of a worldwide *H. pylori* genome collection**. Genome assemblies of *H. pylori* were acquired from the Enterobase database[75]. An additional 63 isolates from the hpAfrica1 and hpNEAfrica population[76] were sequenced on an Illumina MiSeq (2 x 300bp) and assembled with spades 3.15.4 using the –careful and –only-assembler parameters[77] in order to complete the collection. Phylogeographical population assignments were obtained from previous population genetic studies[70,71,76,78]. Sequences with quality (>1000 ambiguous bases) and assembly (>100 contigs) issues were discarded. Based on the *H. pylori* MLST scheme[79], closely related strains were identified (>3 identical MLST alleles) and discarded. The 541 genomes selected and analyzed for this study are listed in Supplementary Data 2, including their phylogeographical population.

**Type I, II(M/G), and III RM system gene sequences**. Genes modulating target-sequence specificity (i.e., genes containing the TRD region) in *H. pylori* were collected from the REBASE database[80]. Depending on the type of RM systems, either the methyltransferase (type II and type III), the specificity subunit (type I and some type IIG), or the RM fusion (type IIG) were selected. The 96 genes analyzed in this study are listed in Supplementary Data 1, including their enzymatic characteristics. All RM systems analyzed in this study are encoded on the chromosome of *H. pylori*. The activities of 88 *H. pylori* methyltransferases analyzed in this study have been previously validated with PacBio Single Molecule Real-Time (SMRT) sequencing data in at least one *H. pylori* strain as indicated in Supplementary Data 1. Among the motifs not validated by PacBio data, two m5c motifs were validated by bisulfite sequencing, four motifs were validated by different methods (see additional details in Supplementary Data 1 and on REbase in corresponding *H. pylori* strains). The methylation of two motifs, GTCTC and CRTANNNNNNNTAG, has not yet been validated experimentally in *H. pylori* and has only been inferred by homology with methylases from other species.

**Distribution of RM system genes in the genome collection**. The genome collection was annotated using the *Helicobacter pylori* genus and species database from Prokka v1.14.5[81] and GNU parallel[81]. Homologs of the RM system genes were searched with the megablast algorithm implemented in BLAST + 2.12.0[82] against a database built with annotated coding sequences (CDS). BLAST hits on single CDS with above 80% nucleotide identity (or 90% for genes undergoing

domain movement) and 70% query coverage were considered positive. BLAST hits below 80% nucleotide identity, 70% query coverage, or including fragmented CDS were considered negative. The frequency for each gene was calculated for both the entire complete collection and for each geographical population. Results were compared to frequencies using blastp and amino acid sequences, instead of blastn and nucleotide sequences, and similar results were obtained (Supplementary Data 6). The frequency in each population was represented as a heatmap with the pheatmap 1.0.12 R package and clustered on both axes using the average-linkage clustering method. Variability between each population was tested using Pearson's chi-square test with Yates' continuity correction.

**Analysis of target motifs frequencies and genomic patterns.** Target-sequence motifs were detected individually in each genome using the Biostrings 2.58.0 Bioconductor R package. For paired methyltransferases that recognize either the exact same motif (M1/M2.HpyPU010X) or complementary non-palindromic motifs (M1/M2.Hpy66X, M1/M2.HpyAVI, and M1/M2.HpyAII), only the motif targeted by the first enzyme (i.e., M1) was considered. The frequency of each motif was scaled to the length of each genome to obtain a normalized motif/Mb unit and plotted either as a bar chart or as a heatmap with the pheatmap 1.0.12R package according to the $Z$-score calculated individually for each motif. The heatmap data was clustered on both axes using the average-linkage clustering method. Variability between each population was tested using Kruskal–Wallis rank sum test (non-parametric one-way analysis of variance). Expected frequencies and compositional bias for each motif were calculated using the methods of Burge and co-authors[46], Pevzner and co-authors[83], and a maximum-order Markov chain model implemented in CBcalc[45]. Additionally, a core gene alignment was created using Roary[84] with -i 80 and -cd 90 parameters and used to calculate the motif frequency in the core genome.

A consensus genome alignment was created by mapping assemblies onto the reference strain BCM300 (RefSeq NZ_LT837687) with BWA 0.7.17[85] (bwa mem with default parameters) and generating consensus sequences with bcftools 1.15.1[86]. Uncovered regions of the reference sequence were masked using the genomecov and substract tools from bedtools 2.30.0[87]. Coverage and pairwise identity data are available for each strain included in the consensus alignment in Supplementary Data 7. Pattern stability was determined by calculating the average number of motifs shared between all pairs of genomes (i.e., motifs located at the same position) in the consensus alignment and reporting it as a proportion of the mean number of motifs per genome. The consensus genome alignment was also used to produce the Venn diagram representing the overlap of ACGT motifs between phylogeographic populations and the gene reading frame analysis of ACGT motifs.

The dependence between methyltransferase frequencies, motif frequencies, pattern stability, and compositional bias was calculated using the distance correlation method implemented in the dcor.test() function from the energy R package. The distance correlation method is a non-parametric test of multivariate independence with the statistical significance evaluated by permutation bootstrap. $p$-values from a right-tailed test are reported.

The interaction between type II methyltransferases and motif densities across geographic populations was determined using a generalized linear model with a quasibinomial distribution and a logit link function, implemented in the glm R package[88]. Goodness-of-it was assessed using a chi-square test ($p < 0.05$) and a pseudo-$R^2$ (McFadden) calculated with the ratio of residual deviance over the null deviance.

**Ancestral-state reconstruction of the ACGT motif patterns.** A random selection of 40 genomes belonging to the main phylogeographical populations hpAfrica1, hpAfrica2, hpAsia2, and hpEastAsia was used as a representative group of *H. pylori* diversity to perform ancestral-state reconstruction analysis. The smaller size of this group compared to the main genome collection ensured a balanced number of genomes per population and reduced the computational complexity of the analysis. A core-genome alignment was created as described above and a phylogenetic tree was produced using iqtree[89] and the TVM + F + R7 substitution model determined by ModelFinder[90]. A time-calibrated tree was generated using the ape R package[91]. Marginal reconstruction of ancestral states was carried out using the stochastic mapping method implemented in the phytools R package[92]. An evolutionary model with fully independent ("all-rates-different") transition rates was selected based on AIC scores and quality of reconstruction. The transition matrix Q was fitted using a continuous-time reversible Markov model (Q = " empirical") and the prior distribution pi on the root of the tree was estimated using the tip character states. Reconstruction was performed for each position containing an ACGT motif in the core-genome alignment. Each allelic variant was considered a distinct state within the reconstruction. The global character-state transition rate matrix was obtained by averaging the transition rates of all individual reconstruction events (the ten most frequent events are displayed in Fig. 6). Gain and loss of ACGT motifs were estimated by first selecting the state with the highest likelihood at each major node of the *H. pylori* tree for each reconstruction (i.e., root and TMRCA nodes of each phylogeographical populations). The selected ancestral states for each node were then added up across all reconstructions and classified into three groups: (1) ACGT motifs, (2) ACAT/ATGT variants (i.e., deamination), and (3) all other allelic variants.

**Statistics and reproducibility.** All data were analyzed using R version 4.1.2 or GraphPad Prism version 7.04. The dependence between methyltransferase frequency, motif frequency, pattern stability, and compositional bias was evaluated across $n = 31$ methyltransferases using the distance correlation method (right-tailed test). The variability of $n = 31$ methyltransferase frequencies between $n = 7$ geographical population of *H. pylori* was tested using Pearson's chi-square test with Yates' continuity correction. The variability of $n = 27$ target motif frequencies between $n = 7$ geographic populations of *H. pylori* was tested using the Kruskal–Wallis rank sum test. The interaction between type II methyltransferases and motif densities across $n = 7$ geographic populations was determined using a generalized linear model with a quasibinomial distribution and a logit link function with the goodness-of-it was assessed using a chi-square test and a pseudo-$R^2$ (McFadden) calculated with the ratio of residual deviance over the null deviance.

**Reporting summary.** Further information on research design is available in the Nature Portfolio Reporting Summary linked to this article.

## Data availability

The dataset supporting the conclusions of this article is available in the NCBI SRA repository, BioProject accession no. PRJNA914092. Source data for the main figures can be found in Supplementary Data 8.

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

## Acknowledgements

We thank Iratxe Estibariz for the early discussions about the variability of methylation patterns and Christine Josenhans for constructive comments on the manuscript. Funding was provided from the Deutsche Forschungsgemeinschaft (project number 158989968-grants SFB900/A1 and SFB900/Z1 to S.S.), by the Bavarian Ministry of Science and the Arts through project HelicoPredict in the framework of the research network bayresq.net, and the German Center for Infection Research (DZIF grants 06.824 and 06.709).

## Author contributions

F.A. and S.S. designed the study. F.A. and W.G. performed the analysis. F.A., W.G., and S.S. interpreted the results. F.A. and S.S. wrote the manuscript. F.A., W.G., and S.S. revised the manuscript. All authors read and approved the final manuscript.

## Funding

## Competing interests

The authors declare no competing interests.
