## [Peer Review File · Communications Biology]

Reviewers' comments:

Reviewer #1 (Remarks to the Author):

This manuscript reports a comprehensive study of the methylome diversity in *Helicobacter pylori*. The authors focus on the characterization of the R-M systems as well as the evolution of methylation patterns, highlighting how the methylome is completely related with the genetic variation and dependent on the phylogeographic structure of the species.

General comments:

In general, the manuscript is interesting and inspiring, an excellent study and discussion about the evolution of methylome in *Helicobacter pylori*. My only (and minor) questions are:

- Although the diversity of RM systems is highly important, I wonder why the authors have not included in their systematic analysis the variability of methyltransferases that lack a cognate restriction endonuclease, the orphan methylases. Which is the inter-strain diversity of orphan methylases? Perhaps, a short paragraph (or figure) that include a briefly summary about this kind of methylases could provide a global vision of all methyltransferases even the most conserved.
- Figure 1. The authors analyze the frequency of 96 genes, but the frequency of 94 targets. Are there some RM systems that methylate/recognize the same motifs?
- Authors do not say anything about the presence of these RM systems in plasmids. Are all encoded on the chromosome?
- Lines 198-200. Authors say: "The diversity of type II RM systems is typically based on the gain and loss of specific systems." An analysis of % identity or % similarity of type II RM systems could be performed. Have you analyzed the number of copies of these genes/genome?

Reviewer #2 (Remarks to the Author):

Aiilond and coworkers present with their study a pioneering work in which they aim to analyze and describe the pan-methylome as a blueprint for the pan-epigenome of *Helicobacter pylori*. Work on genomes, transcriptomes, and proteomes is currently standard, but with this work they demonstrate how to address another Big Data aspect of microbiology, the aforementioned methylome, in analytic fashion.

The study analyzes the presence and distribution of RM systems, as well as the frequency and distribution of Rebase-predicted target motifs. Furthermore, the phylogeographic distribution of genes for RM system components and target motifs is also analyzed. As for the final step, the ancestral-state of the type II m5C motif ACGT was reconstructed.

The study identified 96 different RM systems of types I, II, IIG, and III in 541 genomes, with type II RM systems having the highest mean frequency. Furthermore, the motif density of predicted motifs in this geographically distributed test cohort was also examined, with CATG as the most frequent motif. Several correlations regarding the evolution of RM system genes and target motifs, especially with respect to evolutionary conservation and selection pressure, were extracted.

In my view, the study has one gap, which is due to the genome data set containing only Illumina-generated sequences.

What is missing in my eyes is the complementary analysis of DNA methylation, which can be analyzed, for example, during sequencing using PacBio SMRT sequencing. Measured DNA methylation will correlate with predicted motifs in most cases. Although the number of such currently accessible

datasets/sequences is limited, they should be suitable to correlate the predicted target sequences with the true methylated motifs. In most cases, the percentage methylation rate of the motifs is around 99%, but individual motifs differ significantly in their methylation rate, i.e. the methylation rate is sometimes below 99%. It is difficult to anticipate the outcome of this analysis, but in particular these target sequences, which have a significantly lower methylation rate than the ~99%, could provide complementary insights in terms of abundance among isolates and phylogeographic distribution. The authors should at least try to add such data to their study and, if necessary, discuss them with regard to their limited availability. In all other respects, the manuscript is suitable for publication.

Reviewer #3 (Remarks to the Author):

SUMMARY: I believe this is potentially valuable work but the current presentation has too many problems. The authors used a range of methods to investigate properties of methylation patterns in more than 500 genomes of *Helicobacter pylori*, including 63 isolates sequenced specifically for this study. The manuscript's initial premise is exciting but I was increasingly disappointed as I continued reading. The description of methods frequently lacks important detail, some of the most relevant data is missing, use of statistical methods is problematic, and interpretations are tenuous. I was particularly troubled by the authors' repeated attempts to interpret every real or perceived deviation from randomness as generic evidence for selection without any consideration of alternative explanations.

Specific comments:

1. The method description is rather terse and in parts unclear. For example, I am not sure of the meaning of the sentence "Hits on multiple contiguous CDS, below 80% nucleotide identity or 70% query coverage were considered negative." Providing additional context might be beneficial. In the same paragraph, "Variability between each population was tested using a Chi-square test" does not explain how the test was applied and to what variables, although it becomes clearer later in the results. If providing detailed description of the methods is not possible due to limits on the manuscript length, it might be beneficial to include detailed description of methods as a supplementary file.
2. I was puzzled why the authors compared nucleotide sequences (line 122), presumably using blastn, rather than amino acid sequences (blastp or tblastx). Amino acid sequences diverge more slowly and their alignments tend to be more accurate.
3. I have some concerns about the alignment and identification of ancestral states. The authors state that they constructed core genome alignments by mapping reads to a reference genome using BWA. BWA is intended for alignment of nearly identical sequences, and it is particularly sensitive to insertions and deletions. I could not even find any information about the BWA parameters they used for the alignment; default parameters may not be the most appropriate for this particular purpose. The accuracy of the alignment can subsequently affect the identification of ancestral states. I do not know to what extent potential errors in identification of orthologous ACGT sites can influence the results but I believe that the possible effect of alignment errors should be discussed and, if necessary, the robustness of the phylogenetic analysis with respect to possible alignment errors should be evaluated.
4. Figure 6B includes one transversion in addition to transitions. I would therefore recommend using a different term than transition in describing nucleotide substitutions in the ACGT sites. I am also not entirely clear about the meaning of figure 6A, particularly what is meant by "overlapping motifs". In addition, it is unclear how the authors differentiate between complementary patterns, such as ATGC and ACAT. These refer to the same double-stranded DNA segment and only differ by the strand from which the sequence is read. In coding sequences, one might define the direction with respect to sense

or antisense strand, but it is unclear how the strand should be defined for noncoding sequences. In any case, the distinction between complementary versions of the patterns with one substitution should be explained. If the methyltransferases methylate both strands, it may be appropriate to combine the counts of complementary substitutions, especially if the distinction between the two strands relies on arbitrary decisions.

5. Table 2 does not show sufficient information to support the speculation that flanking direct repeats led to deletions of the methyltransferase genes. I cannot see any repeats in the sequences. Why are the central parts of the sequences not shown? I would like to see the whole sequences of the repeats, where they are located relative to the genes, whether the repeat sequences are identical or only similar, and how conserved they are among different strains. It might also be beneficial to mark the deletions on a phylogenetic tree built from complete genomes (or core set of proteins) to assess whether the deletions occurred multiple times in the evolution.

6. Differentiating between selection and context-dependent or otherwise biased substitution rates is always problematic. The normalization of motif counts by genome length could be insufficient (see, e.g., the Burge et al. 1992, cited in the manuscript). Even if overall nucleotide and oligonucleotide composition of different strains is similar, presence or absence of compositionally atypical segments in some strains could possibly influence the average frequency of the motifs. One possibility to reduce such potential bias is to compare the frequencies in core genome, consisting of regions present in all genomes. If the two approaches lead to different results, that could be potentially informative with respect to evolution of the methylome.

7. The statement about the correlation on line 223 appears to contradict the information provided in the Reporting Summary form by failing to specify what statistical test was used and whether the p-value is one-tailed or two-tailed. Specification of the statistical test and type of p-value is similarly omitted in other instances in the manuscript.

8. One common problem related to interpretation of p-values relates to violations of assumption of independence of trials (observations), which is intrinsic to (almost) all commonly used statistical tests. This almost certainly applies to the data in this manuscript because closely related genomes can hardly be considered independent in the statistical sense. Violations of independence can result in low p-values even if the null hypothesis is correct. This makes me skeptical about interpretations of all statistical tests presented in the manuscript. Moreover, there are often alternative mechanisms that could lead to deviations from the null hypothesis, which are not discussed in the text.

9. In CB calculations, the Karlin method (Burge et al.) and Markov method can produce very different results, and it is not unambiguously clear which one is more appropriate. I would recommend using both methods. Karlin and coworkers also recommended the cutoffs 0.78 and 1.23 for overrepresented and underrepresented oligonucleotides, respectively, which were derived from analysis of actual data. It is certainly unjustified to refer to every value <1 as underrepresented and >1 as overrepresented – values close to 1 are neither. Moreover, even a strong under- or overrepresentation is not necessarily an indication of natural selection. For example, the strong underrepresentation of CpG dinucleotide in human genome is thought to be primarily driven by context-dependent mutations rather than selection. The claim about selection leading to removal of the underrepresented motifs is not supported by the present data.

10. The use of the term “negative selection” is potentially misleading because it is routinely used as a synonym for purifying selection, whereas here the authors mean selection that eliminates certain sequence motifs from the genome, which is not necessarily purifying selection.

11. There is no information about the clustering method used in Fig. 3 and no assessment of cluster stability. Consequently, it is difficult to make any conclusions from the clustering.

12. In the section on lines 251-308 and, to a lesser extent, the following section, the authors repeatedly resort to selection as their explanation for observed relationships among variables. However, a proof of causality requires eliminating all alternative explanations, which the authors did not do. For motif frequencies, the most likely alternative explanation is the effect of context-dependent mutations, although other explanations might also exist. For the geographic distributions of

methyltransferases, possible alternative explanations could include horizontal transfer among strains, lack of divergence, and possibly others. Some of these scenarios could probably be excluded with additional data analysis.

13. The part on lines 338-347 demonstrates the need to differentiate between sense and antisense strand. It is stated that ACGT mostly starts at the second codon position and that C->T or G->A substitutions are mostly synonymous, but a G->A substitution in an ACGT starting at the second codon position would be at the first codon position and it would always be nonsynonymous. More clarity is needed.

14. On lines, 348-364, the authors discuss differences in methylation-dependent mutations in the ACGT motif among different lineages of *H. pylori* strains but never seem to ask, or at least do not show, which strains contain the M.Hpy99XI methylase. This is important because the M.Hpy99XI methylase is presumably responsible for the methylation and therefore one might expect the mutations to occur more frequently in strains that have M.Hpy99XI. This could be expanded into a general comparison between the methylation target frequency in the genome and presence/absence of a methylase targeting that site – this is done in Figure 1 but only in aggregate over all genomes, which could fail to identify unexpected discrepancies in a single genome or a specific lineage.

15. In relationship to the previous comment, one of the most important results of this work would be a table showing the distribution of various methylases among different isolates, but I cannot find it.

16. I would strongly recommend including explanatory legends in the supplementary tables and figures.

Reviewer #1 (Remarks to the Author):

This manuscript reports a comprehensive study of the methylome diversity in *Helicobacter pylori*. The authors focus on the characterization of the R-M systems as well as the evolution of methylation patterns, highlighting how the methylome is completely related with the genetic variation and dependent on the phylogeographic structure of the species.

In general, the manuscript is interesting and inspiring, an excellent study and discussion about the evolution of methylome in *Helicobacter pylori*. My only (and minor) questions are:

- Although the diversity of RM systems is highly important, I wonder why the authors have not included in their systematic analysis the variability of methyltransferases that lack a cognate restriction endonuclease, the orphan methylases. Which is the inter-strain diversity of orphan methylases? Perhaps, a short paragraph (or figure) that include a briefly summary about this kind of methylases could provide a global vision of all methyltransferases even the most conserved.

We are grateful to the reviewer for this positive assessment of our manuscript and the helpful suggestions. To the extent of our knowledge and the information available on the REBASE database, only a single methyltransferase, M.HpyAVII (ATTA^{m6}AT), is systematically orphan in *H. pylori*. Furthermore, we previously characterized a conserved MTase, M.Hpy99III (G^{m5}CGC) and showed that it is orphan in specific phylogeographic populations of *H. pylori* (hpAsia2 and hpEastAsia in particular) but we did not observe any correlation with the evolution of the target motif in our current study. The M.HpyI MTase (C^{m6}ATG) is also known to be orphan in only a subset of the population (i.e. “iceA2” type). Other MTases might also be “facultative orphans” following spontaneous inactivation of the endonuclease through various loss-of-function mutations. Some of our preliminary analysis revealed that such effect is not trivial to predict computationally and thus would require experimental validation of endonuclease activity of all strains. Consequently, orphan methyltransferases were not globally included in the scope of this study. As mentioned in the discussion, M.HpyI and M.Hpy99III are also the only two fully conserved MTases in *H. pylori* and also have the most abundant target motifs in the genome. The presence of endonucleases in type II RM systems has been proposed to be responsible for a decrease (i.e. under-representation) in the number of associated target motifs in other organisms. Consequently, we performed an additional correlation analysis between the presence of the endonuclease and the motif density for both the HpyIP (CATG) and Hpy99IIIP (GCGC) endonucleases (see new Supplemental Figure S6). Interestingly, the absence of either the GCGC or CATG endonucleases in specific geographic populations did not appear to be significantly associated with a higher motif density. This data has been included in the results and discussion section.

- Figure 1. The authors analyze the frequency of 96 genes, but the frequency of 94 targets. Are there some RM systems that methylate/recognize the same motifs?

This is correct. Some RM systems, typically paired in an operon, recognize either the exact same motif (M1/M2.HpyPU010X) or the opposite strands of non-palindromic motifs (M1/M2.Hpy66X, M1/M2.HpyAVI and M1/M2.HpyAII). In our analysis, the targets are typically counted on both strands and thus such complementary non-palindromic motifs have the same number of motifs in the genome. However, paired methylases targeting reverse-complement motifs are not consistently annotated in the REBASE files. For example, M1.Hpy66X and M2.Hpy66X are both annotated as methylating CCATC but M2.Hpy66X actually targets GATGG. This is not the case for M1/M2.HpyAVI and M1/M2.HpyAII which are annotated with their respective complementary motifs. To improve the consistency of our results and avoid duplicated data points, we filtered out results for the second complementary motifs for paired M1/M2 methylases. Consequently, the total number of targets included is now 92. Figure 1B, 2, 4 and Supplemental table 2 have been changed accordingly.

- Authors do not say anything about the presence of these RM systems in plasmids. Are all encoded on the chromosome?

While plasmids are sometimes found in *H. pylori*, they have not been extensively characterized. To the extent of our knowledge, personal observations, and the information available in the REBASE database, all RM systems

analyzed in this study are encoded on the chromosome. Details have been added in the second section of the Methods.

- Lines 198-200. Authors say: "The diversity of type II RM systems is typically based on the gain and loss of specific systems." An analysis of % identity or % similarity of type II RM systems could be performed. Have you analyzed the number of copies of these genes/genome?

Based on the literature on RM systems in *H. pylori* and our BLAST results, RM systems in *H. pylori* are present as single copies. We performed an analysis of % nucleotide identity for all RM systems (Supplemental Figure 2A). Type I and II were overall more conserved than type IIG and III (Fig. S2B). Moreover, the frequencies and % identities of type II methyltransferases were moderately correlated (Fig. S2C).

Reviewer #2 (Remarks to the Author):

Ailloud and coworkers present with their study a pioneering work in which they aim to analyze and describe the pan-methylome as a blueprint for the pan-epigenome of *Helicobacter pylori*. Work on genomes, transcriptomes, and proteomes is currently standard, but with this work they demonstrate how to address another Big Data aspect of microbiology, the aforementioned methylome, in analytic fashion. The study analyzes the presence and distribution of RM systems, as well as the frequency and distribution of Rebase-predicted target motifs. Furthermore, the phylogeographic distribution of genes for RM system components and target motifs is also analyzed. As for the final step, the ancestral-state of the type II m5C motif ACGT was reconstructed. The study identified 96 different RM systems of types I, II, IIG, and III in 541 genomes, with type II RM systems having the highest mean frequency. Furthermore, the motif density of predicted motifs in this geographically distributed test cohort was also examined, with CATG as the most frequent motif. Several correlations regarding the evolution of RM system genes and target motifs, especially with respect to evolutionary conservation and selection pressure, were extracted. In my view, the study has one gap, which is due to the genome data set containing only Illumina-generated sequences. What is missing in my eyes is the complementary analysis of DNA methylation, which can be analyzed, for example, during sequencing using PacBio SMRT sequencing. Measured DNA methylation will correlate with predicted motifs in most cases. Although the number of such currently accessible datasets/sequences is limited, they should be suitable to correlate the predicted target sequences with the true methylated motifs. In most cases, the percentage methylation rate of the motifs is around 99%, but individual motifs differ significantly in their methylation rate, i.e. the methylation rate is sometimes below 99%. It is difficult to anticipate the outcome of this analysis, but in particular these target sequences, which have a significantly lower methylation rate than the ~99%, could provide complementary insights in terms of abundance among isolates and phylogeographic distribution. The authors should at least try to add such data to their study and, if necessary, discuss them with regard to their limited availability. In all other respects, the manuscript is suitable for publication.

We thank the reviewer for the positive assessment of our manuscript which we highly appreciate. Validating the DNA methylation of target motifs is indeed very relevant and has not been mentioned in the manuscript. We apologized for this, we were apparently too close to the work done on *H. pylori* MTases and PacBio sequencing so that we took it for granted. Our group characterized the first two methylomes in *H. pylori* using PacBio SMRT sequencing combined with Tet conversion (DOI: 10.1093/nar/gkt1201) and there are currently 61 *H. pylori* strains analyzed by SMRT sequencing on the REbase database. To provide additional support to our results, we revised Supplemental Table 2 and added, for each motif, reference strains in which the motif has been validated (REbase organism number) and the average percentage of methylation detected in these strains. Among the 96 motifs considered in this study, 88 have been validated with PacBio data in at least one *H. pylori* strain. Among the motifs not validated by PacBio, two m5c motifs were validated by bisulfite sequencing, four motifs were validated by different methods (see additional details in Table S2 and on REBASE in corresponding *H. pylori* strains). Two motifs, GTCTC and CRTANNNNNNTAG, have not been validated experimentally in *H. pylori* and have only been inferred by homology with methylases from other species.

Based on PacBio data, the methylation rates differ between motifs and three motifs with a methylation rate below 90% were observed. However, we believe that the methylation rate is hard to interpret in this specific

context. On the technical side, the methylation status of bases predicted as 'not methylated' has not been extensively validated and thus could be caused by variations in sequencing depth and quality. The amount of PacBio data currently available is also unfortunately not substantial enough to draw correlations between methylation rate variations and strain origin. Moreover, methylation prediction from PacBio are typically provided as an average for a given motif and not for individual motifs across the genome. Comparing methylation patterns of partially methylated motifs across strains could help differentiate between sequencing artefacts and biological effects. Consequently, we decide to not include methylation rate data in our current manuscript.

Reviewer #3 (Remarks to the Author):

SUMMARY: I believe this is potentially valuable work but the current presentation has too many problems. The authors used a range of methods to investigate properties of methylation patterns in more than 500 genomes of *Helicobacter pylori*, including 63 isolates sequenced specifically for this study. The manuscript's initial premise is exciting but I was increasingly disappointed as I continued reading. The description of methods frequently lacks important detail, some of the most relevant data is missing, use of statistical methods is problematic, and interpretations are tenuous. I was particularly troubled by the authors' repeated attempts to interpret every real or perceived deviation from randomness as generic evidence for selection without any consideration of alternative explanations.

1. The method description is rather terse and in parts unclear. For example, I am not sure of the meaning of the sentence "Hits on multiple contiguous CDS, below 80% nucleotide identity or 70% query coverage were considered negative." Providing additional context might be beneficial. In the same paragraph, "Variability between each population was tested using a Chi-square test" does not explain how the test was applied and to what variables, although it becomes clearer later in the results. If providing detailed description of the methods is not possible due to limits on the manuscript length, it might be beneficial to include detailed description of methods as a supplementary file.

We thank the reviewer for the critical assessment of the manuscript and the detailed constructive comments. We believe that we could address them comprehensively and provide the following point by point response. We improved the description at several locations in the methods. In particular we added technical details for the use of BWA, bedtools, Prokka, BLAST, Illumina sequencing, spades and clustering methods. Additionally, we clarified our classification for positive and negative BLAST hits. The description of the correlation analysis between type II methyltransferases and motif frequency across population was also omitted from the methods section and has now been included. Details were also added to every statistical test when necessary (i.e. one or two-tailed, assumptions, multiple-testing correction).

2. I was puzzled why the authors compared nucleotide sequences (line 122), presumably using blastn, rather than amino acid sequences (blastp or tblxastx). Amino acid sequences diverge more slowly and their alignments tend to be more accurate.

It is indeed typically easier to align amino acids sequence due to their lower divergence. Nevertheless, DNA alignments/blastn have been sufficient to infer basic homology of *H. pylori* sequences in the context of our analysis. Furthermore, results were manually inspected to control for any potential alignment issue. Nevertheless, we replicated the homolog search using amino acid sequences and blastp. No major differences in methyltransferases frequencies within our dataset were observed when using nucleotide sequences versus amino sequences. An average absolute difference of 0.2% for type I, 0.3% for type II, 3.5% for type IIG, 1% for type III was observed. The frequencies obtained with both methods have been added as a Supplemental Table S7.

3. I have some concerns about the alignment and identification of ancestral states. The authors state that they constructed core genome alignments by mapping reads to a reference genome using BWA. BWA is intended for alignment of nearly identical sequences, and it is particularly sensitive to insertions and deletions. I could not even find any information about the BWA parameters they used for the alignment; default parameters may not be the most appropriate for this particular purpose. The accuracy of the alignment can subsequently affect the

identification of ancestral states. I do not know to what extent potential errors in identification of orthologous ACGT sites can influence the results but I believe that the possible effect of alignment errors should be discussed and, if necessary, the robustness of the phylogenetic analysis with respect to possible alignment errors should be evaluated.

We agree with the reviewer that particular attention has to be given to the quality of alignments for this type of analysis. While *H. pylori* is a bacterial species with a high degree of sequence variation, unrelated isolates still retain enough identity for BWA to properly align homologous regions (e.g. a recent collaborative study in *H. pylori* using BWA-MEM within Snippy <https://doi.org/10.1038/s41467-022-34475-3>). We typically inspect alignments individually to assess their quality and identify potential issues. Furthermore, we added a Supplemental Table (Table S5) describing the alignment metrics for each strain. Pairwise identities fall within what would be expected between unrelated strains of *H. pylori* and mean coverage percentages show a 90% coverage of our reference on average. These values suggest that a good accuracy of this alignment. Moreover, the agreement between our phylogeny and the literature further confirm that our analysis is accurate.

4. Figure 6B includes one transversion in addition to transitions. I would therefore recommend using a different term than transition in describing nucleotide substitutions in the ACGT sites. I am also not entirely clear about the meaning of figure 6A, particularly what is meant by “overlapping motifs”. In addition, it is unclear how the authors differentiate between complementary patterns, such as ATGC and ACAT. These refer to the same double-stranded DNA segment and only differ by the strand from which the sequence is read. In coding sequences, one might define the direction with respect to sense or antisense strand, but it is unclear how the strand should be defined for noncoding sequences. In any case, the distinction between complementary versions of the patterns with one substitution should be explained. If the methyltransferases methylate both strands, it may be appropriate to combine the counts of complementary substitutions, especially if the distinction between the two strands relies on arbitrary decisions.

In Fig. 6B, transition refers to “character-state transition” rather than a specific type of mutation. Details have been added for clarity. In Fig. 6A, overlapping motifs refer to individual motifs found at the same genome position in distinct strains. For this analysis, all genomes are compared in a pairwise fashion to determine how many motifs are present at the same genomic coordinates in each pair and the average between populations is presented in the Venn diagram. Details have been added for clarity.

Complementary patterns (e.g ATGT/ACAT) are handled differently depending on the analysis. The ancestral reconstruction analysis is performed on the whole genome (i.e. on one strand) and thus each pattern is shown separately (for example on Fig 6B). However, in combination with comment #13, we realized that we did not include some relevant intermediate data concerning the distribution of ACAT and ATGT mutations in the sense and antisense strands of coding sequences. As explained in comment #13, the effect on protein sequences of deamination events happening on ACGT motifs located in the second frame of CDS is completely dependent on the strand (i.e. A^mCGT deamination on the sense strand will produce ATGT while deamination affecting the antisense strand will result in ACAT in the sense strand). Consequently, we modified Fig. 6D to include this information and show that the majority of deamination events happen on the sense strand and second frame of CDS and thus introduce synonymous mutations.

5. Table 2 does not show sufficient information to support the speculation that flanking direct repeats led to deletions of the methyltransferase genes. I cannot see any repeats in the sequences. Why are the central parts of the sequences not shown? I would like to see the whole sequences of the repeats, where they are located relative to the genes, whether the repeat sequences are identical or only similar, and how conserved they are among different strains. It might also be beneficial to mark the deletions on a phylogenetic tree built from complete genomes (or core set of proteins) to assess whether the deletions occurred multiple times in the evolution.

The information about repeats in Table 2 was indeed a bit too minimal (for space reasons, only single partial sequences of repeats were displayed). We created a new supplemental figure (Figure S3) that shows where the repeats are located relative to the RM system genes as well as the remaining single copies of repeats following recombination in strains where the genes are not found. In addition, we added the complete sequence of the

repeats in a Supplemental table (Table S4). We also added the inter- and intra-strain identity of the repeats as well as their prevalence in Table 2.

6. Differentiating between selection and context-dependent or otherwise biased substitution rates is always problematic. The normalization of motif counts by genome length could be insufficient (see, e.g., the Burge et al. 1992, cited in the manuscript). Even if overall nucleotide and oligonucleotide composition of different strains is similar, presence or absence of compositionally atypical segments in some strains could possibly influence the average frequency of the motifs. One possibility to reduce such potential bias is to compare the frequencies in core genome, consisting of regions present in all genomes. If the two approaches lead to different results, that could be potentially informative with respect to evolution of the methylome.

We generated a core gene alignment and used it to calculate motif frequencies again (Fig. S4A). In general, motif frequencies were slightly higher in whole genomes compared to core genomes (Fig. S4B). Seven motifs showed a frequency increase of >25% in the whole versus core comparisons. These motifs have a fairly low frequency overall (min: 91 motif/Mb, max 915 motif /Mb) and thus the differences in absolute number of motifs is small (92 motifs/Mb difference on average for these 7 motifs). In conclusion, it appears that the frequency of the majority of motifs is influenced by the evolution of the whole genome rather than motifs gained or lost through the accessory genome.

7. The statement about the correlation on line 223 appears to contradict the information provided in the Reporting Summary form by failing to specify what statistical test was used and whether the p-value is one-tailed or two-tailed. Specification of the statistical test and type of p-value is similarly omitted in other instances in the manuscript.

Details about the right-tailed permutation test used to evaluate the statistical significance of the distance correlation measurement has been added. Similar details about statistical tests have been added throughout the manuscript.

8. One common problem related to interpretation of p-values relates to violations of assumption of independence of trials (observations), which is intrinsic to (almost) all commonly used statistical tests. This almost certainly applies to the data in this manuscript because closely related genomes can hardly be considered independent in the statistical sense. Violations of independence can result in low p-values even if the null hypothesis is correct. This makes me skeptical about interpretations of all statistical tests presented in the manuscript. Moreover, there are often alternative mechanisms that could lead to deviations from the null hypothesis, which are not discussed in the text.

We do not agree that the genomes analyzed in this study can be considered closely related. All are from separate distinct individuals, and their pairwise genetic relationships support that, since there are no pairs of highly related genomes (e.g., with less than 4% sequence divergence, see new supplemental Table S5). Furthermore, we took particular care to filter related genomes that might come from related individuals within families using MLST haplotypes. Due to its mutation rate and unique environmental conditions existing within the gastric niche of each individual, every *H. pylori* strain is considered to be virtually unique (as depicted by the fact that strains from unrelated individuals almost never share the same MLST sequence type)(see <https://doi.org/10.1093/femsre/fuaa042>). Moreover, the genomes analyzed here cover the worldwide diversity of the species with divergence time up to 60,000 years (see DOI: [10.1371/journal.ppat.1002693](https://doi.org/10.1371/journal.ppat.1002693)). Finally, statistical tests are often used to compare bacterial isolates from the same species (see doi: [10.1038/s41467-023-37302-5](https://doi.org/10.1038/s41467-023-37302-5) ; DOI: [10.1038/s41564-022-01238-1](https://doi.org/10.1038/s41564-022-01238-1) ; doi: [10.1099/acmi.0.000286](https://doi.org/10.1099/acmi.0.000286))

9. In CB calculations, the Karlin method (Burge et al.) and Markov method can produce very different results, and it is not unambiguously clear which one is more appropriate. I would recommend using both methods. Karlin and coworkers also recommended the cutoffs 0.78 and 1.23 for overrepresented and underrepresented oligonucleotides, respectively, which were derived from analysis of actual data. It is certainly unjustified to refer to every value <1 as underrepresented and >1 as overrepresented – values close to 1 are neither. Moreover, even a strong under- or overrepresentation is not necessarily an indication of natural selection. For example, the strong underrepresentation of CpG dinucleotide in human genome is thought to be primarily driven by context-

dependent mutations rather than selection. The claim about selection leading to removal of the underrepresented motifs is not supported by the present data.

We agree with the reviewer that the various methods to calculate compositional bias can produce different results and none of them is universally considered a gold standard. Nevertheless, we chose to use the Karlin method based on a comparative study using thousands of prokaryotic genomes from various species and simulated data which found this method to be the most accurate in this context (see <https://doi.org/10.1134/S0006297918020050>). Based on the other methods considered in this study, we analyzed our dataset using the maximum-order Markov method and the method of Pevzner and co-authors (DOI: [10.1080/07391102.1989.10506528](https://doi.org/10.1080/07391102.1989.10506528)). This data is summarized in a new supplemental figure (Fig S5) and described in the result section. As expected, some variation between methods was observed. However, the main observation that most type II motifs are under-represented in *H. pylori* remain unchanged when considering all three methods and specific cut-offs. The type II motif GTCTC was only found over-represented using the Karlin method and thus is not considered as such anymore. The result section has been modified accordingly.

While it is true that a strong under-representation is not necessarily the result of selective pressures, the currently leading hypothesis to explain under-representation of type II motifs is the motif avoidance hypothesis. According to this hypothesis, type II motifs can progressively get eliminated from the genome to avoid self-restriction by the corresponding endonuclease (see <https://doi.org/10.3389/fmicb.2015.00528> and <https://doi.org/10.1093/gbe/evab097> for example). In this context, self-restriction definitely represents a selection pressure. The same mechanism is observed in the genome of phages targeted by type II RM systems with the elimination of targeted motifs being positively selected.

10. The use of the term “negative selection” is potentially misleading because it is routinely used as a synonym for purifying selection, whereas here the authors mean selection that eliminates certain sequence motifs from the genome, which is not necessarily purifying selection.

We agree with the reviewer that “negative selection” can potentially be misunderstood and the use of this specific term was actually debated when writing the manuscript. Because the potential selection pressures mentioned in the text are already described as being related to the elimination of certain motifs from the genome, we changed all occurrences of “negative selection” to simply “selection” or “selection pressures”

11. There is no information about the clustering method used in Fig. 3 and no assessment of cluster stability. Consequently, it is difficult to make any conclusions from the clustering.

Information about the clustering method for Fig. 3 and Fig. 4 has been added to the method section. Clustering stability has not been assessed since the clusters themselves are not directly used in any downstream analysis and are mostly used as an indication of the categories that can be visually deduced from the heatmaps.

12. In the section on lines 251-308 and, to a lesser extent, the following section, the authors repeatedly resort to selection as their explanation for observed relationships among variables. However, a proof of causality requires eliminating all alternative explanations, which the authors did not do. For motif frequencies, the most likely alternative explanation is the effect of context-dependent mutations, although other explanations might also exist. For the geographic distributions of methyltransferases, possible alternative explanations could include horizontal transfer among strains, lack of divergence, and possibly others. Some of these scenarios could probably be excluded with additional data analysis.

A definite proof of natural selection is not trivial to obtain computationally, in particular in a highly diverse panmictic organism such as *H. pylori*. Nevertheless, we modified the 2nd result section mentioned by the reviewer in order to more accurately reflect what the relationships between motif densities, pattern stability and compositional bias actually suggest. As pointed in comment #10, selection pressures related to the gain or loss of motifs across the genome cannot be easily described in simple terms such as “positive”, “negative”, “diversifying” or “purifying” selection. In the same way, our analysis cannot easily prove that the effects we observed are due to selection and our analysis merely suggest that our results altogether suggest this might be the case.

In this study, motif avoidance and deamination were found to be major sources for the variation in motif frequencies. Both could be considered context-dependent mutations: motif avoidance only affects sequences which are also targeted by an endonuclease present in the genome and prone to self-restriction; deamination only affects 5-methylcytosine. Nevertheless, context-dependent mutation and natural selection are not incompatible. As argued in comment #9, the susceptibility of specific motifs to self-restriction definitely represents a selection pressure. However, differences between type II motifs could be explained by distinct susceptibility to self-restriction, or the existence of other mechanisms counter-acting motif avoidance. These alternative explanations have been added to the discussion. Deamination by itself is purely a biochemical phenomenon that affect random 5-methylcytosines throughout the genome. However, the frequency of m5c motifs is not similarly affected between different motifs or different populations. For example, we have shown previously that the GCGC motif is affected by deamination (Estibariz et al.) but here we show that it does not result in either under-representation, pattern instability or population specific mutation rate. On the contrary, this was the case for the ACGT motif. Although no evidence supports this hypothesis, it is theoretically possible that environmental factors that varies between geographical areas could influence the rate of deamination affecting *H. pylori* locally. In general, population bottlenecks happening both at a large scale during human migration events, or at small scale during within-host evolution could also have an influence on the frequency of context-dependent mutations.

An analysis of methyltransferase divergence has been added in reply to R1 comment #4. *H. pylori* is a naturally competent bacterium and thus horizontal gene transfer can indeed always play a role. However, recombination is so frequent in *H. pylori* that signs of admixture between virtually all geographic groups have been found based on the analysis of populations structure (Linz et al. 2007, Moodley et al. 2012, Thorell et al. 2017, Thorpe et al. 2022 etc). Moreover, as mentioned in result section 3 and in the discussion, the fact that the clustering of all methyltransferase frequencies (Fig. 3 y-axis) mimics the species phylogeny suggest indeed that it is influenced by other mechanisms than selection, such as geographic isolation and genetic drift. Nevertheless, because of the free recombination existing *H. pylori*, combine with the ability of some methyltransferase to undergo intragenic recombination conversion, the maintenance of some methyltransferase genes at very different frequencies in diverse populations suggest some selective effect, in particular when considering the divergence time of each population. For example, the frequency of M.Hpy66VIII cannot be easily explained by other mechanisms even when taking account founder effects, genetic drift, free recombination and human migration patterns. While many scenarios can explain separately the variability of motif and methyltransferases frequencies, the direct correlation demonstrated in the 3rd result section represents solid evidence that the contraction or expansion of motif patterns can be directly influenced by the presence/absence of methyltransferases and thus their methylation status.

13. The part on lines 338-347 demonstrates the need to differentiate between sense and antisense strand. It is stated that ACGT mostly starts at the second codon position and that C->T or G->A substitutions are mostly synonymous, but a G->A substitution in an ACGT starting at the second codon position would be at the first codon position and it would always be nonsynonymous. More clarity is needed.

See reply to comment #4.

14. On lines, 348-364, the authors discuss differences in methylation-dependent mutations in the ACGT motif among different lineages of *H. pylori* strains but never seem to ask, or at least do not show, which strains contain the M.Hpy99XI methylase. This is important because the M.Hpy99XI methylase is presumably responsible for the methylation and therefore one might expect the mutations to occur more frequently in strains that have M.Hpy99XI. This could be expanded into a general comparison between the methylation target frequency in the genome and presence/absence of a methylase targeting that site – this is done in Figure 1 but only in aggregate over all genomes, which could fail to identify unexpected discrepancies in a single genome or a specific lineage.

The frequency of the M.Hpy99XI is definitely an important value and has not been explicitly mentioned in this result section. However, such comparison is performed, at the lineage (geographic population) levels, in the third result section but was not found not significant for M.Hpy99XI and ACGT (see Table S3). To improve this part, we created a new supplemental figure (Fig S7) and provided additional information about the frequency of this

methyltransferase in this result section. The raw presence/absence data has also been provided following comment #15.

15. In relationship to the previous comment, one of the most important results of this work would be a table showing the distribution of various methylases among different isolates, but I cannot find it.

A table showing the distribution of various methylases among different isolates is a very good suggestion. This had been added as a Supplemental table (Table S3) for all strains and all methyltransferases tested in this study.

16. I would strongly recommend including explanatory legends in the supplementary tables and figures.

Explanatory legends for supplementary materials are indeed necessary and has now been added.

Reviewers' comments:

Reviewer #1 (Remarks to the Author):

The authors have addressed accurately all points and comments suggested by the three reviewers. All these suggestions have improved significantly the manuscript and the quality of this paper has increased notably.

Reviewer #2 (Remarks to the Author):

In my opinion, the authors have satisfactorily addressed both my concerns and the comments of the other reviewers. They have made the necessary amendments and improvements to the manuscript, making it suitable for publication.

Reviewer #3 (Remarks to the Author):

The authors addressed most of my recommendations and included additional information I requested. However, one major issue remains. The authors admitted in their response that "a definite proof of natural selection is not trivial to obtain computationally" and that "our analysis cannot easily prove that the effects we observed are due to selection and our analysis merely suggest that our results altogether suggest this might be the case", yet in the manuscript they continue making unqualified claims about natural selection as the only possible conclusion from their results. For example, in the title ("Methylome evolution through lineage-dependent selection in the gastric pathogen *Helicobacter pylori*"), in the abstract ("natural selection shapes both the abundance and distribution of methyltransferases and their specific recognition sequences"), in the Introduction ("we show that type II RM systems are the most conserved in this species and that type II motifs are differentially affected by natural selection"), heading on page 7 ("Methylation patterns are differentially affected by natural selection"), heading on page 8 ("Type II methyltransferases have direct positive or negative selective effects on methylation patterns"), and a number of other statements in the text – selection is only one possible explanation and the data presented in this manuscript provide no indication that it is more likely than other explanations.

The authors' rebuttal suggests that they chose selection because it was previously proposed, not because it is supported by their data: "While it is true that a strong under-representation is not necessarily the result of selective pressures, the currently leading hypothesis to explain under-representation of type II motifs is the motif avoidance hypothesis." The word "selection" should not be used in the title, and the abstract and text of the manuscript need to be modified to express the uncertainties of the interpretations of the results in similar terms that the authors used in the rebuttal. Also note that "motif avoidance" on its own does not necessarily require selection; for example, CG is strongly avoided in the human genome (and many other genomes), which is thought to be a result of context-dependent mutations (methylation-deamination scenario) rather than selection. There are also many exceptions to palindrome avoidance (e.g., CCGG is the most over-represented tetramer in at least some *H. pylori* strains, with Karlin's ratio >1.4), which suggest that other factors might be at play.

In addition to continued unsupported claims related to selection, the author's response to one of my comments indicates a possible misunderstanding related to the concept of statistical independence. Two events, A and B, are considered independent if knowledge that one event occurred does not change the probability of the other event. That is, the events A and B are independent if and only if $\Pr(A|B)=\Pr(A)$ and $\Pr(B|A)=\Pr(B)$. [This condition is sometimes written as $\Pr(A \text{ and } B)=\Pr(A)\Pr(B)$, which is equivalent.] For example, event A could be defined as "an ortholog of some gene X is present

in genome A" and event B as "an ortholog of the same gene X is present in genome B". Gene content, oligonucleotide compositions, and many other genome characteristics tend to be strongly dependent among different strains of the same species, and applications of statistical tests for selection can lead to misleading perception of statistical significance unless properly benchmarked. dN/dS ratio is an example of such benchmarking, where dS represents the background substitution rate in the absence of selection (ideally) or at least at low selection pressure. With respect to the present work, when the authors compare, for example, the motif frequencies among different genomes, the results could be benchmark relative to an analogous comparison of other (similar) oligonucleotides not considered motifs – if those exhibit similar relationships then it is unlikely that those relationships are a consequence of selection that is expected to be specific to the motifs. Something similar could possibly be used to benchmark possible indications of selection related to geographic distribution of the strains.

We thank the reviewers for reevaluating our revised version.

Reviewer 1

Thank you very much for your comments and suggestions which have definitely helped us to improve the paper.

Reviewer 2

Thank you very much for your comments and suggestions which have definitely helped us to improve the paper.

Reviewer 3

Thank you for your comments and for recognizing that most of your concerns have been addressed. As for the additional comments, we have now incorporated more neutral wording and alternative hypothesis in selected parts of the manuscript. In particular:

- Neutral stances have been added for the broad analysis of methyltransferase frequencies and motif densities
- The combined effects founder effect/population bottlenecks, genetic drift in addition to a potential role of selection to explain variation of methyltransferase frequencies in specific cases has been reworded more clearly
- The potential role of context-dependent mutations to explain the evolution of methylation patterns (in addition to motif avoidance) has been introduced. Nevertheless, we pointed out that because all motifs are not similarly affected by such context-dependent mutations, it suggests the existence of additional factors influencing each motif differently (similarly than motif avoidance).
- The interpretation of the correlation analysis has not been modified as we do believe that such significant effects represent a strong evidence that selective effects are at play. Methylation markers represent a plausible biological mechanism to explain the selection of specific motifs considering the multiple phenotypes that have been associated with methylation in *H. pylori* and other bacterial species. In particular, neither motif avoidance or context-dependent mutations can be related to the positive correlation of type II methyltransferase frequencies with m6a motif densities.
- The interpretation related to the evolution of ACGT motifs has been reworded. In particular, we clarified the point that because ACGT can evolve mainly through silent mutations, any hypothetical selection pressure that would act on this motif would be driven by changes in methylation status rather than an effect on protein sequences. Nevertheless, because ACGT and GCGC (both m5c motifs affected by context-dependent deamination) display completely different evolutionary patterns, additional factors, such as selection, must be involved at some level (in either motif).

Finally, we would like to clarify our stance about proofs of natural selection (i.e. rebuttal point 12) as it appears to have been taken out of context. We believe it is rarely possible to provide an absolute, definite, proof of selection with only statistical approaches considering that i) no approach is universally accepted and ii) most methods require some arbitrary assumptions (population size, population structure, mutation rate, recombination rate, etc.). However, we do believe, as argued above, that some of our results represent strong evidence that natural selection is involved in the evolution of specific motifs in relation to their methylation status, and we therefore would very much prefer to leave the title of the paper as it is.